# Multi-AGV path planning method for the textile industry based on improved ant colony optimization algorithm and R-tree line-segment bilayer conflict detection

Wei Xie[1], Xiangle Zheng[1]*, Jiachen Ma[1], Jun Chen[2], Bin Du[1], Xiaoli Wang[1]

1 School of Information Science and Engineering, Harbin Institute of Technology at WeiHai, Weihai, China
2 School of Ocean Engineering, Harbin Institute of Technology at Weihai, Weihai, China

* 18929649382@163.com

## Abstract

The Automated Guided Vehicle (AGV) is crucial for intelligent transportation in textile workshops. Existing centralized system planning research on multi-AGV conflict detection is still limited to simple node or line segment detection, making it difficult to meet the needs of large-scale simultaneous AGV operation to quickly detect whether a conflict exists. Aiming at this problem, this paper designs an R-tree Line-segment Bilayer Conflict Detection mechanism (RLBCD) for efficient collision detection among multi-AGVs by introducing R-tree indexing and analyzing AGV path characteristics, and proposes an Improved Ant Colony Optimization algorithm (IACO) to enhance the quality of AGV path planning, with time as the objective for multi-AGV priority and path strategy design, which ultimately results in collision-free path planning for multiple AGVs. By setting up multiple sets of experiments, it is proved that the RLBCD is not only able to identify the common types of AGV conflicts but also reduces the times of conflict detection comparisons by at least 88.2% and 78.1% compared to the Conflict Based Search and the Grid Time-window Conflict Detection algorithm; the proposed overall algorithm is 29.59 s faster and reduces the cumulative turning angle by 450° compared with the baseline multi-AGV path-planning algorithm, confirming its effectiveness.

## Introduction

With the accelerated transformation of the textile industry to an intelligent production mode, the Automated Guided Vehicle (AGV), as the core carrier of the flexible logistics system, plays a key role in raw material conveying, semi-finished product transfer, and finished product sorting [1]. However, the textile workshop is generally characterized by a dense equipment layout and narrow channels, and multi-AGV operation is very easy to cause deadlock and efficiency loss due to path conflict, which is especially worse when the cluster scale is enlarged. Therefore, in multi-AGV operation collaboration, how to quickly identify whether there is a conflict between AGVs, how to resolve the conflict, and how to plan a collision-free path are the focus of current research by scholars [2]. Currently, there are two popular approaches in multi-AGV collision-free

**Data availability statement:** All relevant data are within the paper and its Supporting Information files.

**Funding:** This work was supported by the National Key R&D Program of China (No. 2022YFB4700601 to W.X.), the National Key R&D Program of China (No. 2022YFB4700602 to W.X.), the Taishan Scholars Foundation of Shandong Province (No. tsqn201909153 to X.W.), the Ministry of Education Industry-University Cooperative Education Project (No. 22086429092517 to W.X.).

**Competing interests:** The authors have declared that no competing interests exist.

domain planning: a distributed reaction resolution approach, where AGVs dynamically resolve conflicts that arise during their movement towards the target location [3–6], and a centralized system planning approach, where collision-free paths are computed before AGVs start moving towards the target location.

The research focus of this paper is on centralized system planning methods. Many centralized system solutions are currently popular, the Conflict Based Search (CBS) achieves collision-free planning via a two-level decomposition: at the lower level, optimal single-agent paths are computed independently without constraints from other agents; at the upper level, upon detecting spatiotemporal occupancy at the vertex or edge level, the corresponding spatiotemporal constraints are introduced and re-planning is triggered, thereby progressively eliminating conflicts [7–10]. CBS can find a globally optimal solution that minimizes the total path cost of all AGVs; however, its drawbacks are evident: because it must compare vertex and edge conflicts across agents, the upper-level branching on constraints and the lower-level re-planning grow increasingly expensive as the number of AGVs and the environmental complexity increase. In parallel to CBS, a line of work based on spatio-temporal networks/time-window search explicitly discretizes time and builds a time-expanded graph, internalizing collision-avoidance constraints into the network via capacity limits and forbidden edges, and then solves the problem using shortest-path or minimum-cost flow methods. This approach is highly verifiable but suffers from rapid state-space growth when the time granularity is fine, the task horizon is long, or the fleet size is large [11,12].Another strand emphasizes regional mutual exclusion: by imposing exclusive occupancy or reservation on bottleneck corridors and intersection areas to preclude passing encounters and concurrent conflicts, it offers a simple engineering implementation; its limitation is that poorly chosen interlock-region granularity induces a trade-off between conservatism and throughput [13,14]. In addition, some studies move conflict suppression upstream to the task-assignment stage by incorporating potential congestion or crossing risks into the cost function and constraints, yielding task–path matches with fewer anticipated conflicts before execution; the drawback is that performance depends on the fidelity of the risk characterization, and local geometric conflicts during operation typically still require online compensation [15].

Beyond the foregoing mature frameworks, another common remedy is to assign AGV priorities and confirm conflicts via a Grid-based Time-window Conflict Detection (GTCD) algorithm. Unlike CBS, GTCD emphasizes priority-based pre-screening: conflicts are detected directly by testing for spatiotemporal overlap on edges, without decomposing into vertex versus edge conflicts or constructing a constraint tree. Any overlap is deemed a conflict and is resolved at runtime through waiting or path re-planning. GTCD is often combined with heuristic path planners such as Particle Swarm Optimization (PSO) [16,17], Genetic Algorithm (GA) [18,19] and Ant Colony Optimization (ACO) [20,21].Guo et al. [22] introduced an ant fallback mechanism based on traditional ACO to enhance the adaptability and used a temporary avoidance-research strategy to solve the opposite conflict in AGV; Zhang et al. [23] incorporated the vehicle waiting time, the distance of the vehicle from the target point,

and the urgency of the task performed by the trolley into the AGV priority consideration, which successfully circumvented various common conflict types of AGVs; Yang et al. [24] proposed elastic time window combined with ACO to improve the efficiency of multi-AGV path planning; Zhong et al. [25] used hybrid GA-PSO for AGV path planning in automated container terminal scenarios to achieve integrated scheduling of multi-AGV collision-free path planning; Pratissoli et al. [26] designed a hierarchical control architecture for multi-AGV path planning based on a time window to effectively reduce the occurrence of collision and deadlock processing accidents among multi-AGVs; Zhu et al. [27] proposed a New Ant Colony Optimization (NACO) algorithm that introduces an a priori time window to dynamically adjust the pheromone concentration, make the path time function the iterative goal of ant colony pheromone update, and add a strategy decision threshold in the path optimization process, which affects the accumulation of ant colony pheromone according to the severity of the conflict situation, so as to carry out the path replanning and waiting strategy selection.

The above centralized system planning algorithms for multi-AGV conflict detection invariably utilize AGV shortest-path line segment or node information to facilitate conflict detection. However, as the AGV scale increases and the operational range expands, the delay in retrieving conflicts significantly increases. This has a detrimental effect on the algorithm's real-time performance. Aiming at this problem, this paper designs an R-tree Line-segment Bilayer Conflict Detection mechanism (RLBCD) for efficient collision detection among multi-AGVs by introducing R-tree indexing and analyzing AGV path characteristics and proposes an Improved Ant Colony Optimization algorithm (IACO) on the basis of NACO [27], which ultimately results in a multi-AGV collision-free path planning method. First, IACO quantifies the path deflection angle into the time cost function by introducing a kinematic constrained turn time model, and amends the pheromone updating mechanism of NACO to improve the quality of AGV path planning; then IACO is run for the AGVs that start at the same time to obtain the optimal time cost of the initial path, and each AGV is prioritized through the before-and-after task assignment and the cost of the initial path, and then according to the priority and initial line segment run RLBCD algorithm, using AGV path characteristics coarse screening to quickly obtain the suspected conflict line segment area, fine screening to confirm whether the conflict really exists, to ensure the accuracy of conflict detection; if there is no conflict, the AGV to the initial optimal path driving. Otherwise, the initial path combines with the conflict detection situation to perform precise time waiting; the high-priority AGV's minimum path segment is mapped as a dynamic obstacle, deeply integrating with IACO to enable collision-free path replanning for the low-priority AGVs. Finally, by measuring the cost of the waiting and path replanning strategy, the smaller time policy is selected to transition from single-AGV path planning to multiple-AGV path synergy, all aimed at prioritizing time. In extreme cases where no deconfliction strategy is feasible for a low-priority AGV at a given time, the algorithm applies a preemptive intervention: the task will be canceled at the current time instant and restarted in the planning for AGVs departing in the next time batch. This strategy is consistent with priority-ordered planning and serves as a feasibility prerequisite for collision-free multi-AGV operation.

Overall, this study develops along the CBS framework, adopting a two-level decomposition of "first compute single-agent optima, then perform global deconfliction": the lower level employs IACO, while the upper level implements deconfliction by combining priority assignments with an RLBCD mechanism. The contributions are summarized as follows:

(i) Propose IACO to improve the path-planning quality of individual AGVs.

(ii) Leverage the priority scheme to map high-priority AGV paths into dynamic-obstacle regions for low-priority AGVs, tightly coupling IACO so that, through the pheromone-positive-feedback mechanism of ant colonies, optimal alternative paths under collision-free constraints are found automatically.

(iii) Introduce an RLBCD mechanism for rapid conflict detection across multiple AGV paths, enhancing detection efficiency while maintaining accuracy.

The subsequent chapters are arranged as follows. The "Materials and Methods" chapter describes the model assumptions of the algorithm proposed in this paper, the IACO algorithm formula used for single-AGV path planning, and the

multi-AGV collision-free path planning algorithms: RLBCD mechanism design, waiting strategy implementation, dynamic obstacle generation, and path replanning. The "Experimental Results and Analysis" chapter initially delineates the experimental parameters of this paper's overall algorithm. Subsequently, several experimental scenarios are constructed: (i) the proposed overall algorithm is evaluated on three common conflict types; (ii) RLBCD, CBS, and GTCD are run on six sets of AGV path instances with varying numbers of paths (i.e., fleet sizes) to compare conflict-detection counts and runtime; (iii) IACO is compared with ACO and NACO for single-AGV path planning; and (iv) end-to-end tests are conducted under simultaneous operation with fleets of different sizes to compare the proposed overall algorithm with a NACO-based multi-AGV path-planning algorithm. Finally, the "Conclusion" chapter summarizes the research of this paper.

## Materials and methods

### Model-based assumptions

(i) To better reflect the realization of multi-AGV cooperative cooperation strategy, the textile workshop map unit grid division (Fig 1), black grid for the static obstacle area, reflecting the textile industry, a variety of raw materials and equipment of different sizes, not marching; white grid can be carried out in the region; red grid for the dynamic obstacle area, only in the special time is not permitted to enter; different AGV travel routes using different color markings; the solid line represents the final conflict-free path, the dotted line represents the original conflicting path, the small circle

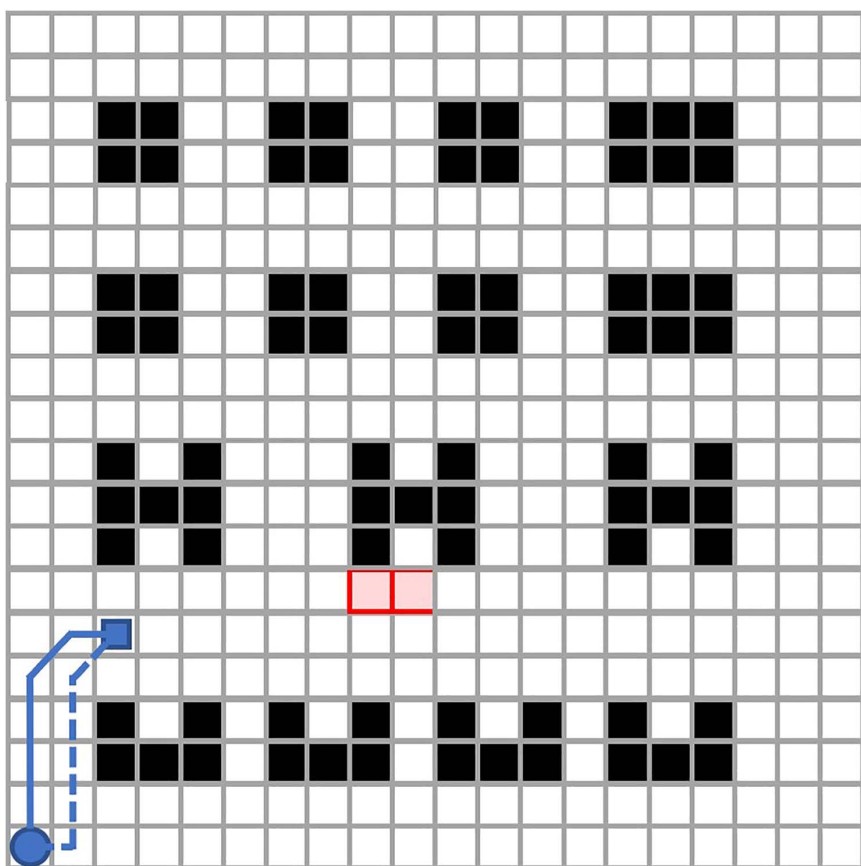

**Fig 1. Textile workshop map.**

represents the AGV starting point, and the small square represents the AGV target end point, and the coordinates of each grid node are $(x, y, t)$, the node set is $N$, and the connection between the nodes is represented by the edge set $E$.

(ii)   It is stipulated that the ant colony walks by moving with the center of each frame of the gridded map as the node target, and the movable directions are the eight domain directions.

(iii)   All AGVs run at the same speed in m/s, assuming that the acceleration during the start phase and the deceleration during the stop phase of the AGV are not taken into account.

(iv)   AGV can realize in-situ turning, the turning time with the turning angle is uniformly increasing, and the turning radius is zero.

(v)   Provided that all AGVs are of the same size and that the map is divided into the same size per grid length as the AGV size, ensuring that only one AGV can exist at a grid node location in m units.

**Single AGV path planning based on IACO**

**State transfer probability.**   The probability $p_{ij}^k$ that ant $k$ moves from node $i$ to node $j$ is defined as:

$$p_{ij}^k = \frac{[\tau_{ij}]^\alpha [\eta_{ij}]^\beta}{\sum_{l \in N_k^{\text{allowed}}} [\tau_{il}]^\alpha [\eta_{il}]^\beta}$$

(1)

- $\alpha$: parameters of pheromone importance
- $\beta$: parameters for the importance of heuristic information
- $N_k^{\text{allowed}}$: the set of ant $k$ currently accessible neighboring nodes
- $\tau_{ij}$: pheromone concentration from node $i$ to node $j$
- $\eta_{ij}$: heuristic information from node $i$ to node $j$

**Heuristic function.**   Euclidean distance is used as the heuristic function for the ACO algorithm. Euclidean distance:

$$D_j = \sqrt[2]{(x_j - x_{\text{end}})^2 + (y_j - y_{\text{end}})^2}$$

(2)

Heuristic function:

$$\eta_{ij} = \frac{1}{D_j + 1}$$

(3)

**Pheromone update time cost function $C_{\text{path}}$:**

1. Define the pheromone update time cost function $C_{\text{path}}^k$:

The traditional ant colony algorithm basically takes the shortest path as the basis of pheromone update, but for AGV planning paths, what should be considered is the time cost of the AGV actually arriving at the destination from the starting point, instead of the simple path length. The reason is that, for the AGV, turning time is costly, which literature [27] is

also concerned about this situation and has introduced the concept of the time function to minimize the number of turns in the AGV driving process. AGV driving process of the number of turns, this approach to a certain extent played a role in the effect, but the disadvantage is that the scholar proposed turn time is not enough to fit with the actual, the loss of time between the turn angle is inconsistent, which to a certain extent affects the degree of Ant Colony Decision Making Path deflection, resulting in many large-angle turns appeared and the debugging of the algorithm process output not clear; we further improve the performance of the algorithm based on the consideration of the motion characteristics of the AGV; the cost of the turn time is subdivided, and the pheromone updating time cost function $C_{path}^{k}$ is defined as the total time of the path taken by the ant $k$:

$$C_{path}^{k} = t_{move} + t_{turn} \tag{4}$$

$$t_{move} = \frac{L_{total}}{v} \tag{5}$$

$$t_{turn} = \sum_{i=1}^{N_{turns}} \theta_i \times t_{turn\_unit}^{IACO} \tag{6}$$

- $t_{move}$: total cost of straight time (s)

- $t_{turn}$: total cost of turning time (s)

- $L_{total}$: total length of path (m)

- $v$: constant speed of the AGV (m/s)

- $t_{turn\_unit}^{IACO}$: turning time per unit angle (s/°)

- $\theta_i$: angle of the ith turn (°)

2. Calculation of turning angle $\theta_i$:

In order to calculate the turning angle, it is necessary to determine the change of direction between three adjacent nodes in the path:

$$\mathbf{v}_1 = P_i - P_{i-1} \tag{7}$$

$$\mathbf{v}_2 = P_{i+1} - P_i \tag{8}$$

- $P_{i-1} = (x_{i-1}, y_{i-1})$: coordinates of the (i-1) node

- $P_i = (x_i, y_i)$: coordinates of the ith node

- $P_{i+1} = (x_{i+1}, y_{i+1})$: coordinates of the (i+1) node

  Calculation of turning angle$\theta_i$:

$$\theta_i = \arccos\left(\frac{\mathbf{v}_1 \cdot \mathbf{v}_2}{\| \mathbf{v}_1 \| \cdot \| \mathbf{v}_2 \|}\right) \times \left(\frac{180}{\pi}\right) \tag{9}$$

- $\mathbf{v}_1 \cdot \mathbf{v}_2$: dot product of vectors

- $\| \ \|$: modulus of a vector

**Pheromone increment function $\Delta\tau_{ij}$:** At the end of each iteration, after each ant completes the path search, the pheromone increment is updated according to the total cost of the path time $\Delta\tau_{ij}$:

$$\Delta\tau_{ij} = \sum_{k=1}^{M} \Delta\tau_{ij}^{k}(T) \tag{10}$$

$$\Delta\tau_{ij}^{k}(T) = \begin{cases} \frac{Q}{C_{path}^{k}(T)}, & \text{ant k passes through edge } (i,j) \\ 0, & \text{other} \end{cases} \tag{11}$$

- $C_{path}^{k}(T)$: total path cost of ant k in Tth iteration

- M: total number of ants

- Q: pheromone intensity

## IACO algorithm pseudo-code

The pseudo-code of the above IACO algorithm is as follows:

**Algorithm 1.  Single AGV path planning based on IACO.**

**Input: Set of start and end points for multiple AGVs**

 Grid-based environment representation

 Set start and end node

**Output:** Optimal path with minimum time cost

1: **for** T = 1 to MaxIterations **do**

2: Place M ants at the start node

3: **for** each ant **do**

4: **while** ant not at end node **do**

5: compute state transition probabilities, $(\text{ant } k \leftarrow p_{ij}^{k})$

6: determine $\theta_i$ using relative direction of neighboring three nodes

7: move to the next node

8: **end while**

9: calculate total time cost of the ant's path, $(\text{ant} k \leftarrow C_{path}^{k}(T))$

10: **end for**

11: **for** each edge $(i,j)$ **do**

12: update pheromone based on time cost of all ants' paths

13: perform pheromone evaporation

14: **end for**

15: **end for**

16: **return** Optimal path with minimum time cost over all iterations

## Multi-AGV path planning strategy

**AGV priority assignment mechanism.** Since the single AGV path planning is using the IACO algorithm, the optimal time path of each AGV can be accurately obtained; for this reason, this paper proposes a time-first multi-AGV priority allocation mechanism on this basis, with the sequence of each AGV executing the task as the first priority judgment, i.e., the priority of AGVs that are already running will be higher than that of those that depart later, and if the AGVs are

departing simultaneously, then run single AGV in parallel to IACO algorithm to determine the priority of the AGVs with the initial path cost; the smaller the time cost, the higher the priority, and if the priority is the same, then take a randomized priority ordering, so as to avoid the cycle as if waiting and falling into deadlock when multiple AGVs have conflicts. When an AGV task is canceled or encounters an accident, the priority bubbling mechanism is used to dynamically adjust the AGV task priority. Assume there are $2n$ AGVs dispatched at two time instants $t1$ and $t2$ ($t1 < t2$), with $n$ AGVs departing at each instant. Then, at time $t2$, the global priority set is

$$A = \{A_{t1}, A_{t2}\} \tag{12}$$

- $A$: represents AGV priority set ordering

- $A_{t1} = \{AGV_1^{t1}, AGV_2^{t1}, \ldots, AGV_n^{t1}\}$ : the priority sequence of AGVs that departed at time $t1$ and are currently en route

- $AGV_{11\sim 1n}$: represents the prioritization of $n$ AGVs departing at the same time based on the path-minimum cost as the first and the stochastic principle as the second.

**RLBCD model.** R-tree is a spatial indexing structure that realizes multi-level indexing of spatial data through Minimum Bounding Rectangle, which can effectively improve the query efficiency of spatial data [28]. In this paper, drawing on the hierarchical screening idea of R-tree, combined with the path characteristics of AGV trajectory planning, a spatio-temporal R-tree line segment bilayer conflict detection mechanism is proposed to quickly identify the potential collision risk between multi-AGV paths. Next, we first introduce several common AGV conflict types and then introduce the specific algorithmic process of RLBCD.

1. Type of AGV conflict

Since the present model assumes that the speeds of the AGVs are all the same, there is no pursuit conflict, and there are only three types of conflicts, i.e., node conflict, opposite conflict and crossover conflict.

As shown in Fig 2(a), a node conflict will occur when AGV1 and AGV2 occupy the same grid simultaneously; as illustrated in Fig 2(b), an opposite conflict occurs when the starting points of AGV1 and AGV2 are at the endpoints of each other; and as depicted in Fig 2(c), a crossover conflict, which is a moving process collision, will happen when the travel paths of AGV1 and AGV2 intersect.

2. RLBCD pre-conditions

Upon completion of the IACO simulation, the initial path nodes and their timestamps for each AGV in the same batch can be obtained. Assume there are $n$ AGVs in this batch. For the $\alpha$-th AGV ($\alpha = 1, 2, \ldots n$), let $N^\alpha$ denote the number of path nodes and $N_{turns}^\alpha$ the number of turning nodes; then the path-node set of AGV$\alpha$ is defined as $\mathcal{D}_0^\alpha$.

$$\mathcal{D}_0^\alpha = \{n_1^\alpha, \ldots, n_{N^\alpha}^\alpha\}, n_1^\alpha = (x_1^\alpha, y_1^\alpha, t_1^\alpha) \tag{13}$$

**(a)** Node conflict **(b)** Opposite conflict **(c)** Crossover conflict

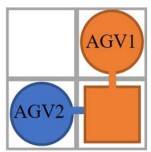 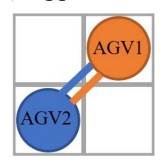 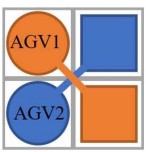

**Fig 2. Type of AGV conflict.**

Based on the turning-coordinate indices, each AGV's path can be partitioned into same-direction segments in ascending order of departure time, yielding the start/end coordinates and the traversed time windows. The set of same-direction segmented paths for AGV$\alpha$ is denoted by $\mathcal{D}_1^\alpha$ as follows:

$$\mathcal{D}_1^\alpha = \{L_1^\alpha, L_2^\alpha, \ldots, L_{N_{\text{turns}}^\alpha+1}^\alpha\}, \quad L_j^\alpha = (P_{s,j}^\alpha, P_{e,j}^\alpha, t_j^\alpha) \tag{14}$$

- $L_j^\alpha$: the $j$-th contiguous same-direction path segment of AGV$\alpha$
- $P_{s,j}^\alpha = \left(x_{s,j}^\alpha, y_{s,j}^\alpha\right)$, $P_{e,j}^\alpha = \left(x_{e,j}^\alpha, y_{e,j}^\alpha\right)$: start and end coordinates of path segment $L_j^\alpha$
- $t_j^\alpha = (t_{s,j}^\alpha, t_{e,j}^\alpha]$: line segment $L_j^\alpha$ time domain (s)
- $t_{e,j}^\alpha = t_{s,j+1}^\alpha$, $P_{e,j}^\alpha = P_{s,j+1}^\alpha$: ensure time continuity and accurate path segmentation

Define the global path-node set $\mathcal{D}_0$ and the global same-direction segmented path set $\mathcal{D}_1$ for all AGVs whose routes have been confirmed. For the AGVs departing in the current batch, if no AGV is already operating on the grid map, then $\mathcal{D}_0$ and $\mathcal{D}_1$ are empty; if some AGVs are already executing tasks, their paths are placed into $\mathcal{D}_0$ and $\mathcal{D}_1$. By the priority definition, AGVs currently executing tasks have higher priority than those departing now, and because path planning proceeds in descending priority order, the global sets $\mathcal{D}_0$ and $\mathcal{D}_1$ can be viewed as the node and path sets of higher-priority AGVs, which constitute the final outputs of the algorithm. If no AGV is executing a prior task, then upon completion of the current batch simulation the following sets are obtained.

$$\mathcal{D}_0 = \{\mathcal{D}_0^1, \ldots \mathcal{D}_0^\alpha \ldots, \mathcal{D}_0^n\}, \quad \mathcal{D}_1 = \{\mathcal{D}_1^1, \ldots \mathcal{D}_1^\alpha \ldots, \mathcal{D}_1^n\} \tag{15}$$

Further, suppose the $j$-th same-direction path segment $L_j^\alpha$ is partitioned into $g_j^\alpha$ elementary line segments by connecting consecutive node pairs. Using the start/end coordinates of $L_j^\alpha$, the relevant node information can be queried from the set $\mathcal{D}_0^\alpha$. We then define the set of subdivided segments for the same-direction path $L_j^\alpha$ of AGV$\alpha$ as follows ($i=1,2, \ldots, g_j^\alpha$):

$$\mathcal{D}_2^\alpha \left(L_j^\alpha\right) = \{l_{j,1}^\alpha, l_{j,2}^\alpha, \ldots, l_{j,g_j^\alpha}^\alpha\}, \ l_{j,i}^\alpha = \left(P_{s,j,i}^\alpha, P_{e,j,i}^\alpha, t_{j,i}^\alpha\right) \tag{16}$$

- $l_{j,i}^\alpha$: the $i$-th subdivided segment of the $j$-th contiguous same-direction path segment of AGV$\alpha$
- $P_{s,j,i}^\alpha = \left(x_{s,j,i}^\alpha, y_{s,j,i}^\alpha\right)$, $P_{e,j,i}^\alpha = \left(x_{e,j,i}^\alpha, y_{e,j,i}^\alpha\right)$: the start and end coordinates of the subdivided segment $l_{j,i}^\alpha$
- $t_{j,i}^\alpha = (t_{s,j,i}^\alpha, t_{e,j,i}^\alpha]$: the time interval of the subdivided segment $l_{j,i}^\alpha$ (s)
- $t_{e,j,i}^\alpha = t_{s,j,i+1}^\alpha$, $P_{e,j,i}^\alpha = P_{s,j,i+1}^\alpha$: ensure time continuity and accurate path segmentation

3. RLBCD coarse screening stage

In the coarse-screening stage, spatiotemporal conflict detection for the current-priority AGVs adopts a binary criterion: (i) whether the time windows overlap, and (ii) whether the minimum distance between segments is below the safety threshold. Segments are examined in the order of the departure times of the current-priority AGV$\alpha$: segments of $\alpha$ are taken from $\mathcal{D}_1^\alpha$, while higher-priority segments are taken from the global set $\mathcal{D}_1$. Let $L_j^\alpha$ be the segment currently under consideration for AGV$\alpha$, and $L_k^\beta$ the segment of a higher-priority AGV$\beta$. The candidate conflict set for $L_j^\alpha$ is defined as:

$$\mathcal{C}_1^\alpha(L_j^\alpha) = \{L_k^\beta \in \mathcal{D}_1 | t_j^\alpha \cap t_k^\beta \neq \varnothing \wedge d_{\min}(L_j^\alpha, L_k^\beta) < D_{\text{safe}}\} \tag{17}$$

- $D_{\text{safe}}$: safe distance thresholds (m)

- $d_{\text{min}}$: minimum distance between line segments $L_j^\alpha, L_k^\beta$ (m)

- $t_j^\alpha \cap t_k^\beta$: means the direction continuous line segment $L_j^\alpha, L_k^\beta$ time domain overlap

- $\varnothing$: empty set

In the coarse-screening stage, we first filter by whether the time windows overlap and then compute exact segment-to-segment distances to obtain $\mathcal{C}_1^\alpha(L_j^\alpha)$. If $\mathcal{C}_1^\alpha(L_j^\alpha) = \varnothing$, then $L_j^\alpha$ is deemed conflict-free and we proceed to the next contiguous segment $L_{j+1}^\alpha$; otherwise, the procedure enters the fine-screening stage.

4. RLBCD fine screening stage

Based on $\mathcal{C}_1^\alpha(L_j^\alpha)$ and $L_j^\alpha$, retrieve the sets $\mathcal{D}_2^\beta(L_k^\beta)$ and $\mathcal{D}_2^\alpha(L_j^\alpha)$, and perform fine-screening for precise conflict determination. The fine-screening proceeds as follows: first, use time-window overlap to quickly filter subdivided segment units $I_{k,i'}^\beta$ and $I_{j,i}^\alpha$; suppose the selected pair is $I_{j,1}^\alpha$ and $I_{k,2}^\beta$. Finally, compute the exact distance between the subdivided segments $I_{j,1}^\alpha$ and $I_{k,2}^\beta$. A conflict is confirmed if and only if the following conditions are satisfied:

$$\text{Real\_conflict}(I_{j,1}^\alpha, I_{k,2}^\beta) = \left(d_{\text{min}}(I_{j,1}^\alpha, I_{k,2}^\beta) < D_{\text{safe}}\right) \cap \left(t_{j,1}^\alpha \cap t_{k,2}^\beta \neq \varnothing\right) \tag{18}$$

After fine screening, the precise conflict-information set $\mathcal{C}_2^\alpha(L_j^\alpha)$ is obtained:

$$\mathcal{C}_2^\alpha(L_j^\alpha) = \left\{(I_{j,1}^\alpha, I_{k,2}^\beta), (I_{j,2}^\alpha, I_{m,1}^\gamma), ....\right\} \tag{19}$$

- $I_{j,1}^\alpha = (P_{s,j,1}^\alpha, P_{e,j,1}^\alpha, t_{j,1}^\alpha)$

- $(I_{j,1}^\alpha, I_{k,2}^\beta)$: a confirmed pair of conflicting subdivided segments between AGV $\alpha$ and higher-priority AGV $\beta$

The conflict pairs in $\mathcal{C}_2^\alpha(L_j^\alpha)$ are ordered chronologically by the subdivided segments of AGV $\alpha$. If one subdivided segment conflicts with multiple higher-priority AGVs, they are ordered by the arrival times of the higher-priority AGVs, with earlier arrivals listed first. Simultaneous arrivals do not occur, because once a preceding AGV successfully applies a deconfliction strategy, the arrival times become distinct.

If the precise conflict-information set $\mathcal{C}_2^\alpha(L_j^\alpha)$ is empty, coarse screening continues on the next variable-length segment $L_{j+1}^\alpha$. If the entire path of AGV $\alpha$ has no spatiotemporal conflicts with higher-priority AGVs, no strategy adjustment is needed; retain $\mathcal{D}_0^\alpha$ and $\mathcal{D}_1^\alpha$, and store them into the global sets $\mathcal{D}_0^\alpha$ and $\mathcal{D}_1^\alpha$, respectively, before proceeding to conflict checking for the next-priority AGV. If $\mathcal{C}_2^\alpha(L_j^\alpha)$ is nonempty, the process enters the subsequent deconfliction-strategy phase.

5. RLBCD time-complexity

In our analysis, no spatial index structure is instantiated; the algorithm operates on the segment–time-window representation directly. We assume constant-time segment-to-segment distance in 2-D and linear matching of time windows, on which the following big-O bounds are derived. Since RLBCD requires the same-direction segment information of higher-priority AGVs for coarse screening, let $N^{\alpha ll}$ denote the total number of nodes belonging to all known higher-priority AGVs together with the current-batch AGVs. The time complexity of the preparation stage is therefore

$$T_{\text{prepare}}(\mathcal{D}_1) = O(N^{\alpha ll}) \tag{20}$$

Assume the total number of segments in the set $\mathcal{D}_1$ is $H$. For the AGV $\alpha$ currently under inspection, the number of same-direction segments is $N_{\text{turns}}^\alpha + 1$. Let the segment being checked be $L_j^\alpha$, whose number of subdivided segments is

$g_j^\alpha$. After coarse screening by time-window matching and exact distance filtering, the candidate set $\mathcal{C}_1^\alpha(L_j^\alpha)$ contains $G_j$ elements. Hence, the coarse-stage complexity for $L_j^\alpha$ depends on the total number $H$ of higher-priority segments, giving

$$T_{\text{coarse}}(L_j^\alpha) = O(H) \tag{21}$$

Entering the fine-screening stage, suppose $G_j$ is further subdivided into $K_j$ elementary segments. Using time-window overlap and exact distance checks, the final number of conflicting subdivided-segment pairs is $Q_j$. The fine-stage complexity is

$$T_{\text{fine}}(L_j^\alpha) = O(K_j + g_j^\alpha + (g_j^\alpha K_j)) = O(g_j^\alpha K_j) \tag{22}$$

- $K_j$, $g_j^\alpha$: subdivision of $G_j$ and $L_j^\alpha$, each costing $O(1)$ per segment

- $(g_j^\alpha K_j)$: pairwise time-window checks between subdivided segments, proportional to the number of checks

  Therefore, for the segment $L_j^\alpha$, the time complexity of one RLBCD execution is

$$T_{\text{RLBCD}}(L_j^\alpha) = O(H + g_j^\alpha K_j) \tag{23}$$

Overall, the time complexity of path conflict detection for AGV$\alpha$ is

$$T_{\text{RLBCD}}(\text{AGV}\alpha) = O\left((N_{\text{turns}}^\alpha + 1)H + \sum_{j=1}^{N_{\text{turns}}^\alpha+1} g_j^\alpha K_j\right) \tag{24}$$

In the coarse-screening stage, if the time-window and distance checks exhibit heavy overlap, then $G_j = H$.

In the fine-screening stage, let the upper bound on the number of subdivisions of any same-direction segment be $g_{max}$. Then $K_j \leq Hg_{max}$ and $g_j^\alpha \leq g_{max}$. The worst case is therefore:

$$T_{\text{coarse}}(L_j^\alpha) = O(H)$$

$$T_{\text{fine}}\left(L_j^\alpha\right) = O\left(g_j^\alpha K_j\right) \leq O\left(Hg_{max}^2\right)$$

$$T_{\text{RLBCD}}\left(L_j^\alpha\right) = O(H) + O\left(Hg_{max}^2\right) = O\left(Hg_{max}^2\right)$$

$$T_{\text{RLBCD}}(\text{AGV}\alpha) = O\left((N_{\text{turns}}^\alpha + 1)Hg_{max}^2\right) \tag{25}$$

The above process is as Fig 3:

**Dynamic waiting strategy design.** If $\mathcal{C}_2^\alpha(L_j^\alpha)$ is not an empty set, then AGV$\alpha$ next tries the waiting strategy first, copies the information about the current AGV path time from the $\mathcal{D}_0^\alpha$ and $\mathcal{D}_1^\alpha$ set to the $\mathcal{D}_3^\alpha$ and $\mathcal{D}_4^\alpha$ set, and according to the $\mathcal{C}_2^\alpha(L_j^\alpha)$ set, finds the first conflicting line segment position that satisfies the condition in chronological order and carries out the waiting time calculation. The minimum waiting time $\Delta t_1$ is calculated:

$$\Delta t_1 = max\left(0, t_e^{high} - t_s^{low}\right) + \Delta t_{\text{buffer}} \tag{26}$$

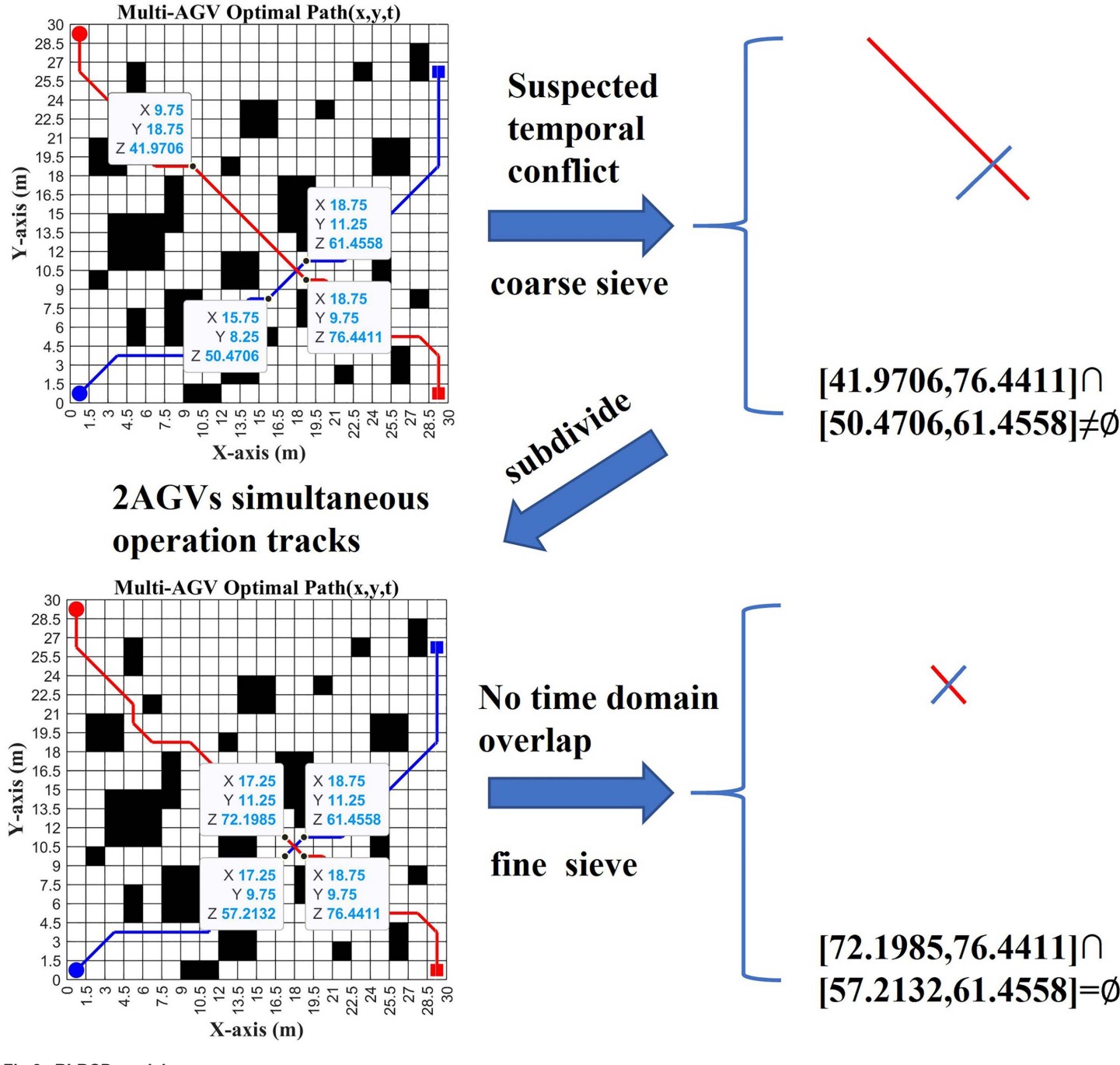

**Fig 3. RLBCD model.**

- $\Delta t_{buffer}$: safety buffer time

- $t_s^{high}$, $t_e^{high}$: start and end time for high priority AGVs passing through conflicting line segment pairs

- $t_s^{low}$, $t_e^{low}$: start and end time for low priority AGVs passing through conflicting line segment pairs

- $t^{low\_new} = (t_s^{low} + \Delta t_1, t_e^{low} + \Delta t_1]$: time domain after update of low priority AGV

After resolving the first conflicting line segment pair, the time domain of AGV$\alpha$'s subsequent path changes due to the addition of the waiting time, update the time domain of AGV$\alpha$ $\mathcal{C}_2^\alpha(L_j^\alpha)$'s subsequent conflicting line segment pairs, and if the originally conflicting time domains no longer overlap, eliminate the conflicting pairs in $\mathcal{C}_2^\alpha(L_j^\alpha)$ until the next conflicting pair is found that can't be eliminated from the overlap, and continue to perform the waiting time $\Delta t_2$ computation until all the $\mathcal{C}_2^\alpha(L_j^\alpha)$ conflicting line segment pairs are gone, update the information of the $L_j^\alpha$ line segments in the set of $\mathcal{D}_3^\alpha$ and $\mathcal{D}_4^\alpha$, and start a new conflict detection for the next segment of unfixed length line segment of AGV$\alpha$. If there is no more subsequent conflict information, then the waiting strategy ends; if there is, then continue to process the new $\mathcal{C}_2^\alpha(L_{j+1}^\alpha)$ set and continue with the waiting time calculation. Assuming that $n$ waiting times have been processed, the waiting strategy ends and the total waiting cost time is calculated:

$$C_{\text{wait}} = C_{\text{original}} + \sum_{k=1}^{n} \Delta t_k \tag{27}$$

- $C_{\text{wait}}$: total waiting time

- $C_{\text{original}}$: AGV1 original path cost time

- $\sum_{k=1}^{n} \Delta t_k$: sum of n waiting times, should be less than $\Delta t_{max}$

- $\Delta t_{max}$: the maximum waiting time allowed to avoid encountering deadlocks and infinite waiting.

Once the cumulative waiting time exceeds $\Delta t_{max}$, then $C_{\text{wait}}$ will be set to be inf. Cancel the waiting strategy and choose the path replanning strategy. If the waiting strategy succeeds, the new $\mathcal{D}_3^\alpha, \mathcal{D}_4^\alpha$ and $C_{\text{wait}}$ are obtained.

**Dynamic obstacle area generation.** To prevent the high-priority obstacle region from being generated repeatedly and inefficiently, assuming that AGV$\alpha$ has finalized its ultimate path, the dynamic-obstacle region $B_\alpha$ (see Fig 4) is generated from the node set $\mathcal{D}_0^\alpha$. Specifically, the region associated with the minimal (elementary) segment traversed by AGV$\alpha$ is expanded—depending on the mode of motion—into two distinct forms of spatiotemporal obstacle. Meanwhile, the dynamic obstacle must be time-bounded: its activation interval is maintained according to the passage time of the higher-priority AGV.

$$B_\alpha = \begin{cases} \{(x_s^\alpha, y_s^\alpha), (x_e^\alpha, y_e^\alpha)\}, [t_s^\alpha, t_e^\alpha] & \text{horizontal/vertical movement} \\ \{(x_s^\alpha, y_s^\alpha), (x_e^\alpha, y_e^\alpha), (x_s^\alpha, y_e^\alpha), (x_e^\alpha, y_s^\alpha)\}, [t_s^\alpha, t_e^\alpha] & \text{slanting movement} \end{cases} \tag{28}$$

- $(x_s^\alpha, y_s^\alpha), (x_e^\alpha, y_e^\alpha)$: start and end coordinates of the AGV$\alpha$ 's minimum path line segment

- $[t_s^\alpha, t_e^\alpha]$: start and end times of the AGV$\alpha$ 's minimum path line segment

Define the global set of dynamic-obstacle regions for higher-priority AGVs as $B = \{B_1, \ldots, B_n\}$.

**Path replanning strategy design.** After the waiting strategy is over, the path replanning strategy is carried out to obtain the dynamic obstacle region information of the high-priority AGV required by the current AGV according to the set $B$, which is input to the IACO algorithm to carry out the path replanning. Similar to the ACO algorithm in doing planning for static obstacles will add the nodes of the obstacle region to the taboo table, leaving only the region nodes that can be selected, if the ant colony arrives near the conflict region under the new path planning, the displaced path cost time is slower or earlier than the original one, thus leading to the disappearance of the original time domain window of the conflict, the dynamic obstacle region should disappear, and if the next time the path generated by the colony still arrives at

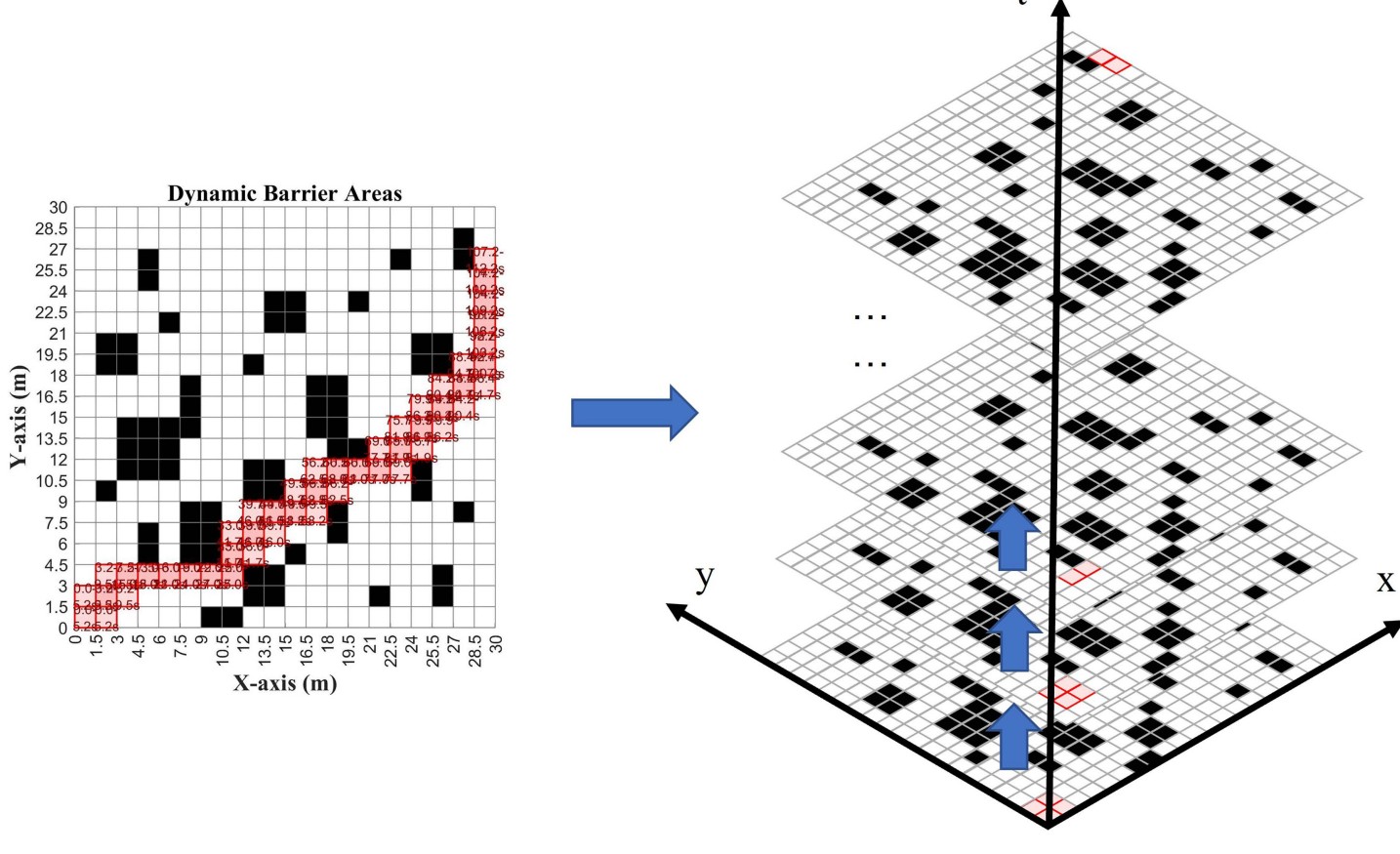

**Fig 4. Dynamic obstacle generation.**

the same general time as the original path, then the dynamic obstacle at this location continues to keep generating, and the nodes in the obstacle region are added to the colony state transfer taboo table, so that the colony can not choose the nodes in this region, and eventually iterates other relatively more optimal paths. If the path replanning is successful, the new path time information will be stored in $\mathcal{D}_0^\alpha$ and $\mathcal{D}_1^\alpha$. If a feasible path cannot be found within the maximum number of replanning attempts, set $C_{\text{replan}} = \inf$ and cancel the path replanning, and choose the waiting strategy.

$$C_{\text{replan}} = t_{\text{new\_move}} + t_{\text{new\_turn}} \tag{29}$$

- $C_{\text{replan}}$: total path replanning time

- $t_{\text{new\_move}}$: new total cost of straight time (s)

- $t_{\text{new\_turn}}$: new total cost of turning time (s)

   **Strategy selection.** Compare $C_{\text{wait}}$ and $C_{\text{replan}}$, whose time cost is high. If $C_{\text{wait}}$ is small, then we need to update the new path information $\mathcal{D}_3^\alpha$ and $\mathcal{D}_4^\alpha$ into $\mathcal{D}_0^\alpha$ and $\mathcal{D}_1^\alpha$. If $C_{\text{wait}}$ is large, do not update $\mathcal{D}1$ and keep the re-planned path information. After the strategy updates to $\mathcal{D}_0^\alpha$ and $\mathcal{D}_1^\alpha$ are finalized, the dynamic-obstacle region $B_\alpha$ for AGV$\alpha$ is generated and inserted into the set $B$, so that it can serve as an obstacle reference when re-planning the path for the next

lower-priority AGV. In extreme cases where no deconfliction strategy is feasible for a lower-priority AGV at a given time, a preemptive intervention is adopted: the task will be canceled at the current time instant and restarted in the planning for AGVs departing in the next time batch, without affecting already-committed high-priority paths. This strategy is consistent with priority-ordered planning and constitutes a feasibility prerequisite for collision-free multi-AGV routing. The pseudo-code of the above overall multi-AGV collision-free path planning algorithm is as follows:

**Algorithm 2.  Multi-AGV Collision-free Path Planning with RLBCD and IACO.**

**Input: Set of start–goal nodes for AGVs departing at time $t1$**

**Output:** Set $\mathcal{D}_0, \mathcal{D}_1, B$

1: Obtain set $\mathcal{D}_0, \mathcal{D}_1, B$

2: $\mathcal{D}_0^\alpha, \mathcal{D}_1^\alpha \leftarrow$ IACO (AGV$\alpha$, $B$)

3: $A_{t1} \leftarrow$ prioritize AGVs based on priority rules

4: **for** AGV$\alpha$ in set $A_{t1}$ order **do**

5:      Conflict $\leftarrow$ RLBCD check with higher-priority AGVs

6:      **if** Conflict exists then

7:        Apply path replanning strategy

8:        Apply dynamic waiting strategy

9:        **if** strategy partially succeeds then

10:          Output successful strategy for AGV$\alpha$

11:          Update $\mathcal{D}_0^\alpha, \mathcal{D}_1^\alpha$

12:          Generate set $B_\alpha$

13:          Update set $\mathcal{D}_0, \mathcal{D}_1, B$

14:        **else**

15:          Cancel AGV$\alpha$'s current task

16:           Insert AGV$\alpha$ into the next time set $A_{t2}$

17:            Re-sort $A_{t1}$

18:        **end if**

19:          Pick the minimum time-cost strategy: wait or replan

20:          Obtain new set $\mathcal{D}_0^\alpha, \mathcal{D}_1^\alpha$

21:          Generate set $B_\alpha$

22:          Update set $\mathcal{D}_0, \mathcal{D}_1, B$

23:      **else**

24:        Retain $\mathcal{D}_0^\alpha, \mathcal{D}_1^\alpha$

25:        Generate set $B_\alpha$

26:        Update set $\mathcal{D}_0, \mathcal{D}_1, B$

27:      **end if**

28: **end for**

29: **return** $\mathcal{D}_0, \mathcal{D}_1, B$

## Experimental results and analysis

### Algorithm parameterization

The experimental scenario used the textile workshop map in Fig 1. The algorithm code was written using a Python program, and the algorithm was run on an Intel Core i9-14900HX 2.20 GHz processor (16.00 GB RAM) with the operating system Windows 11, and the parameters of the experimental algorithm were set as in Table 1.

**Table 1. Algorithm parameterization.**

| Parameter | Setting | Parameter | Setting |
|---|---|---|---|
| $\alpha$ | 1 | M | 50 |
| $\beta$ | 5 | $D_{safe}$/Map grid size | 1m |
| $t_{turn\_unit}^{IACO}$ | 10/180 s/° | $t_{turn\_unit}^{NACO}$ | 1 s/turn |
| $v$ | 0.5m/s | Q | 30 |
| $\Delta t_{buffer}$ | 0.2s | Maximum Iterations | 100 |
| $\Delta t_{max}$ | 30s | Maximum replanning times | 2 |

## Experimental simulation and result analysis

(i) First of all, the algorithm proposed in this paper carries out several common kinds of conflict tests, the use of two AGVs to perform the task at the same time, respectively, and setting up different start and end points, such as Table 2, to achieve different types of conflict.

As shown in Fig 5, the algorithm proposed in this paper has good recognition as well as proper handling of various conflict types, when the conflict area occupies a very small area, such as node conflict, the algorithm will choose the waiting strategy according to the principle of the lowest overall path time cost, and when there are opposite conflicts as well as crossover conflicts, the time cost of choosing the waiting strategy is much higher, so the originally planned dashed old path is discarded and path re-planning is carried out, and a solid new path is obtained. The solid line new path is obtained.

(ii) Then the RLBCD mechanism is tested centrally, compared with GTCD and CBS conflict detection methods. In these experiments, all three methods are given the same initial spatiotemporal path-node data for the AGVs (Fig 6(a)-(f)), obtained by running IACO from the start/end points in Table 3. The concrete implementations are as follows: GTCD adopts the priority scheme proposed in this paper to rank AGVs and then performs direct edge-level spatiotemporal conflict checks (using the shortest-edge criterion). RLBCD further processes the path-node data into same-direction polylines and applies a coarse-to-fine spatiotemporal conflict detection. CBS does not assign priorities; it directly performs minimum-edge and minimum-vertex spatiotemporal conflict checks on each AGV's path-node data. All three methods use a consistent conflict criterion: if the spatial separation is below $D_{safe}$ and the time windows overlap under linear matching, a conflict is declared. The detection count is recorded once per complete spatiotemporal check of a path, with the objective of locating each AGV's first conflict position. Finally, we compute the number of path-information checks and the task completion time for each method to validate the proposed approach.

$$\frac{GTCD - RLBCD}{GTCD} \times 100\% \qquad \frac{CBS - RLBCD}{CBS} \times 100\%$$

**Table 2. Common conflict type settings.**

| Conflict type | Node conflict | | Opposite conflict | | Crossover conflict | |
|---|---|---|---|---|---|---|
| ID | AGV1 | AGV2 | AGV1 | AGV2 | AGV1 | AGV2 |
| Start | (6, 12) | (9, 15) | (6, 11) | (6,20) | (13,20) | (6,20) |
| End | (11,12) | (9,11) | (14, 20) | (14,11) | (9,17) | (10,17) |
| Color | Blue | Red | Blue | Red | Blue | Red |

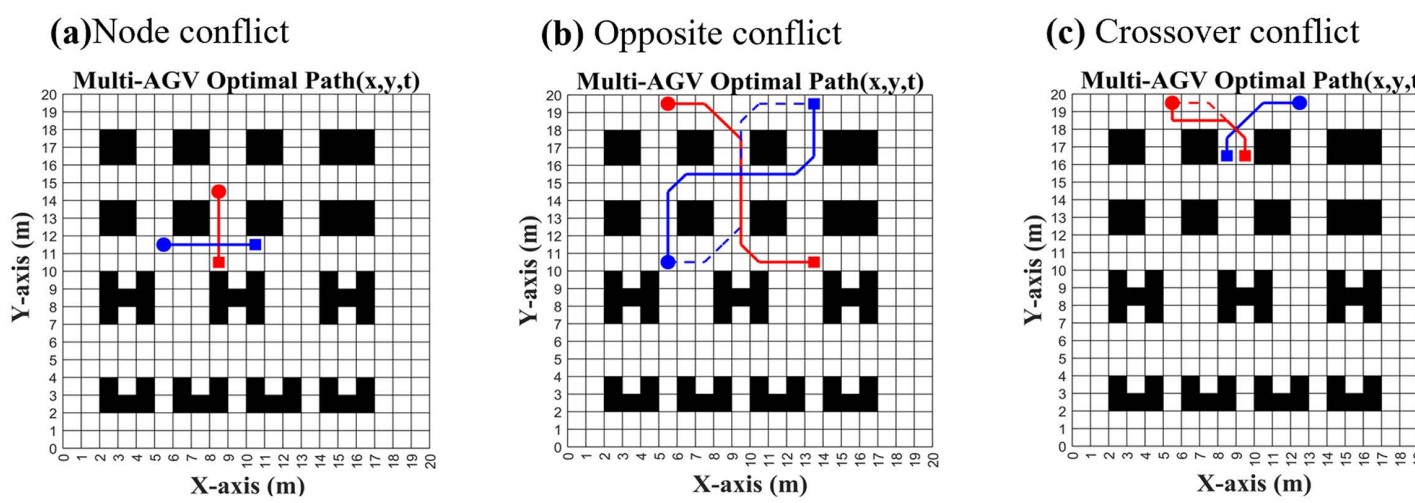

**(d)** Node conflict selection waiting strategy 3D demonstration

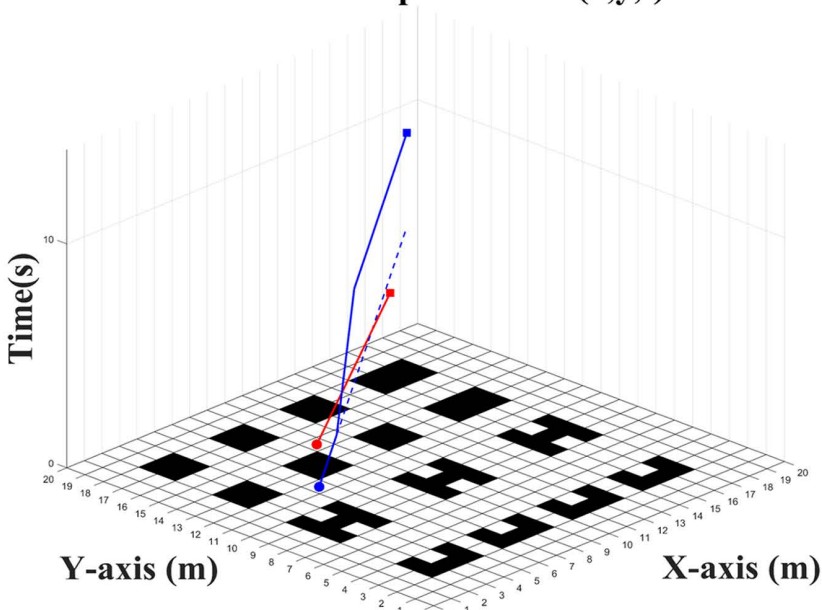

**Fig 5. Simulation of common conflict types.**

As demonstrated in Fig 6(a)-(f), with the gradual increase in the number of AGVs, there is a substantial overlap in path information between AGVs, leading to a notable rise in the probability of collision. As further demonstrated in Fig 7, CBS necessitates the execution of point and edge conflict detection on disparate AGV path information to ensure conflict retrieval accuracy. To obtain the information of the earliest conflict point of each AGV, CBS experiences a parabolic rise in the number of detections with the increase in the number of AGVs, thereby significantly reducing time efficiency.

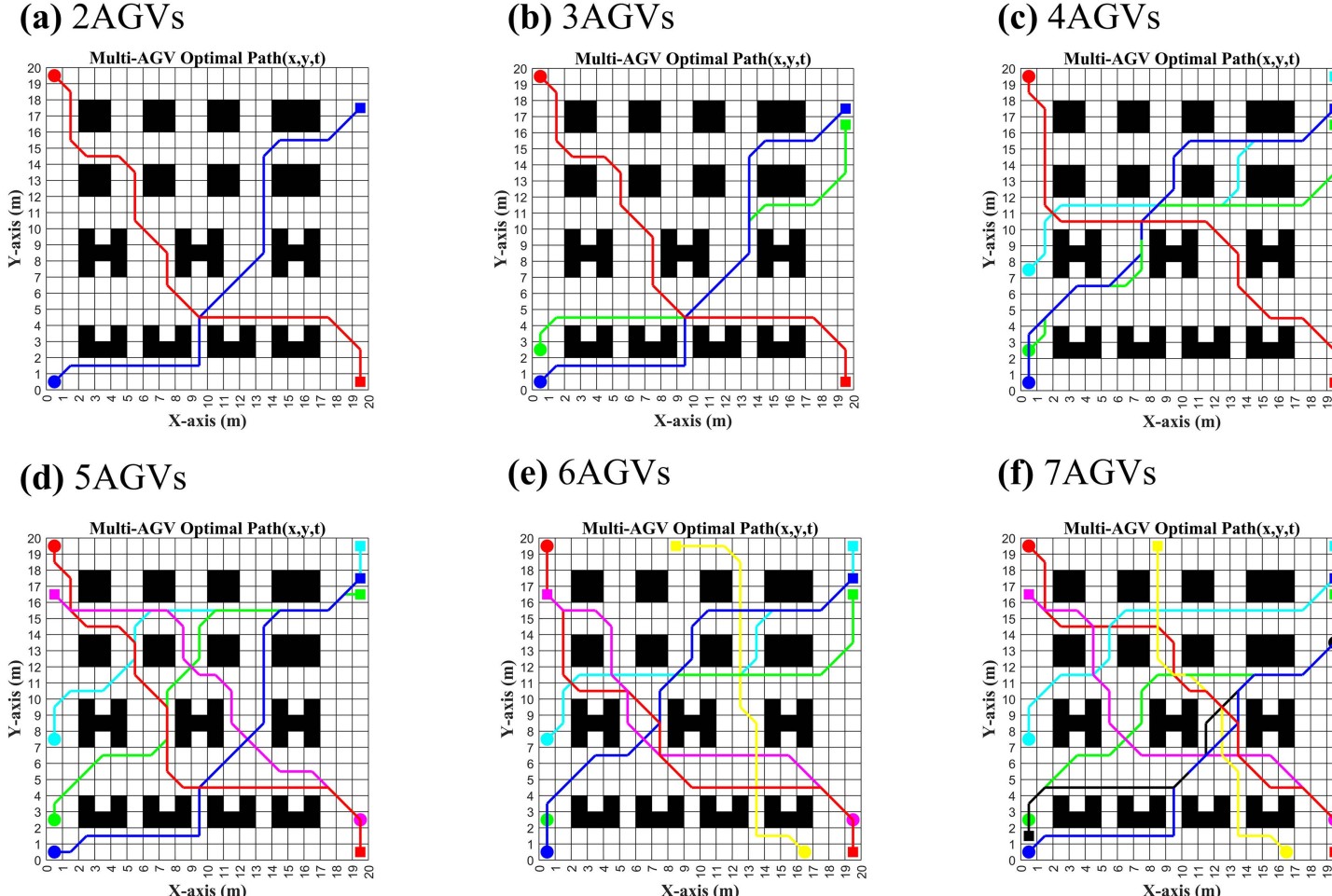

**Fig 6. Spatio-temporal path information for different numbers of AGVs.**

**Table 3. Multi-AGV simultaneous operation settings.**

| Task | 1 | 2 | 3 | 4 | 5 | 6 | 7 |
|---|---|---|---|---|---|---|---|
| ID | AGV1 | AGV2 | AGV3 | AGV4 | AGV5 | AGV6 | AGV7 |
| Start | (1,1) | (20,1) | (3,1) | (8,1) | (3,20) | (1,17) | (14,20) |
| End | (18,20) | (1,20) | (17,20) | (20,20) | (17,1) | (20,9) | (2,1) |
| Color | Blue | Red | Green | Azure | Pink | Yellow | Black |

Conversely, GTCD exhibits a capacity for adaptable augmentation of the retrieval range within the grid time domain by assessing the movement modes of AGVs, thereby circumventing an exhaustive retrieval of global AGV path segment information. Each AGV is required to identify its first and smallest conflicting path segment based on its time before and after its own path information. It does not need to detect subsequent path information, thereby significantly reducing the number of comparisons of path information and enhancing time efficiency. However, when conflicts arise in proximity to the AGV section's termination or when an AGV lacks conflicts with other AGVs, the number of times the AGV's path information is compared remains significant.

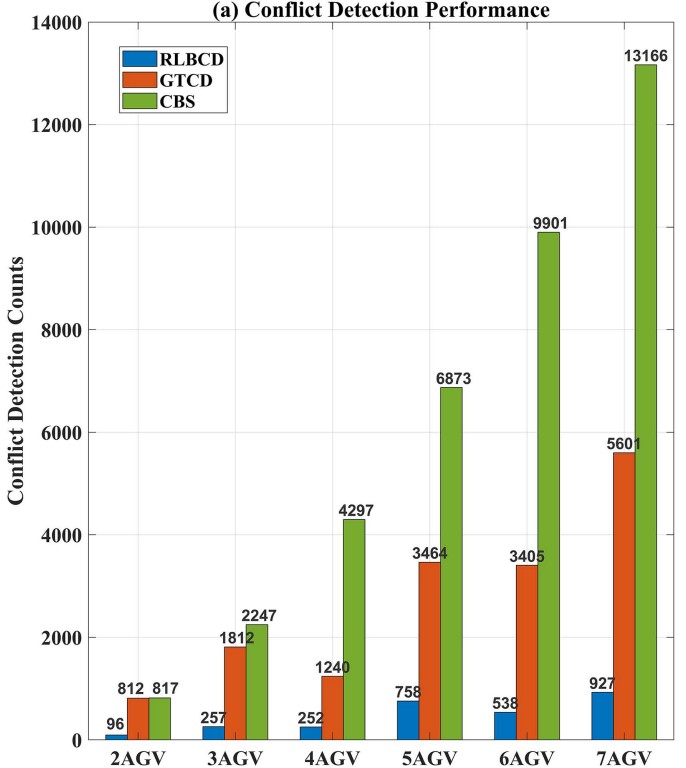
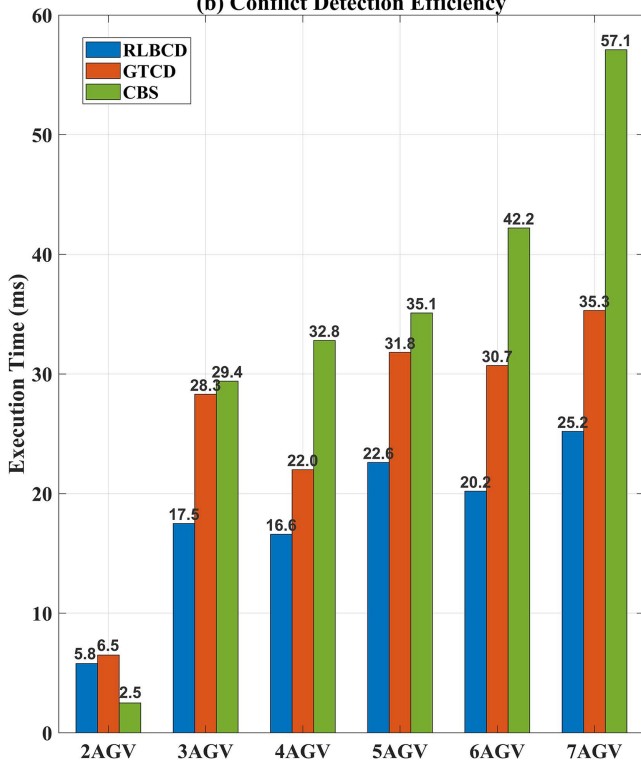

**Fig 7. Simulate different algorithms for detecting different numbers of AGVs.**

As illustrated in Table 4, relative to CBS and GTCD, RLBCD effectively screens out a minimum of 78% of the collision-free line segments through coarse screening under the established experimental conditions. It merely requires refinement of the number of line segments for the remaining 22%. This approach leads to a substantial reduction in the number of comparisons between different AGV path information, while ensuring the accuracy of conflict retrieval and enhancing the efficiency of conflict retrieval. However, it is noteworthy that when the number of AGVs is small and there are only two for conflict detection, the RLBCD, due to its algorithmic structure, is more complex than the CBS and results in a less optimal conflict retrieval time.

(iii)　　Finally, using the AGV task start and end point data in Table 3, the overall algorithm proposed in this paper is compared with the NACO-based multi-AGV collision-free strategy algorithm by setting up a single-AGV versus multi-AGV collision-free path planning test to quantify the task completion time, turning angle, and path planning length to validate the effectiveness of the proposed algorithm. Parameter settings are listed in Table 1; in NACO, the only difference from IACO is the unit turning time: $t_{turn\_unit}^{NACO}$ is count-based, whereas $t_{turn\_unit}^{IACO}$ is angle-based.

$$Ideal\ time\ (ACO) = \frac{Path\ length}{v}, Ideal\ time\ (NACO) = \frac{Path\ length}{v} + t_{turn\_unit}^{NACO} \times Turn\ times$$

$$Actual\ time(ACO/NACO) = \frac{Path\ length}{v} + t_{turn\_unit}^{IACO} \times Turn\ angle$$

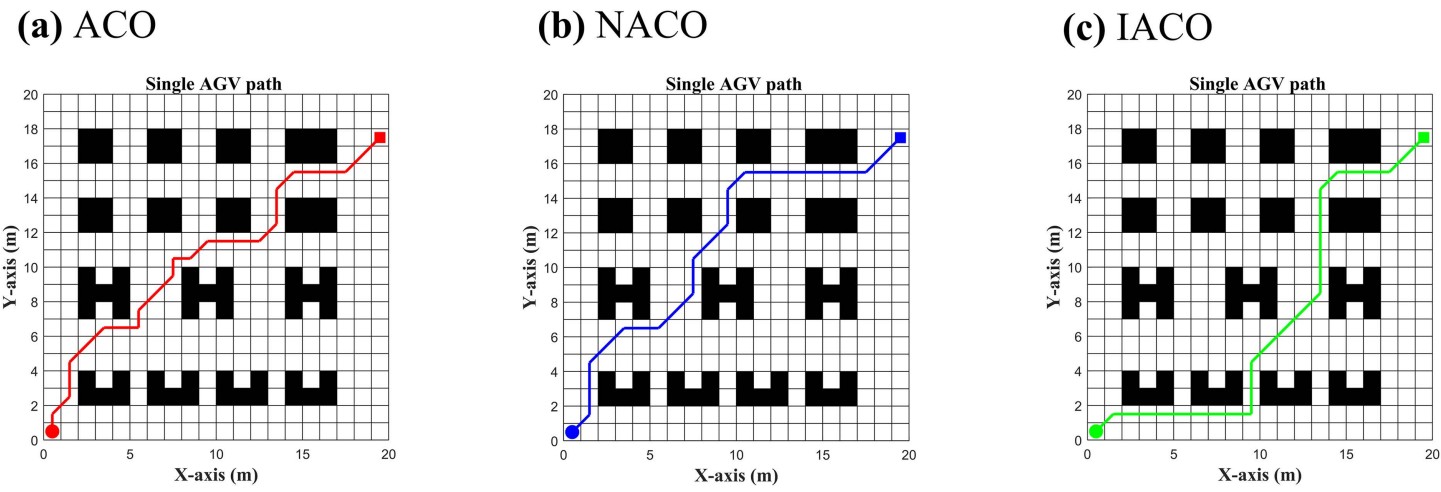

Table 4. Post-processing of different algorithms for conflict detection.

| Indicators | Number of path comparisons | | | | Actual execution time (ms) | | | |
|---|---|---|---|---|---|---|---|---|
| | RLBCD | GTCD | CBS | Improve% | RLBCD | GTCD | CBS | Improve % |
| 2AGV | 96 | 812 | 817 | 88.2/88.2 | 5.8 | 6.5 | 2.5 | 10.8/**-132** |
| 3AGV | 257 | 1812 | 2247 | 85.8/88.6 | 17.5 | 28.3 | 29.4 | 38.2/40.4 |
| 4AVG | 252 | 1240 | 4297 | 79.7/**94.1** | 16.6 | 22 | 32.8 | 24.5/49.4 |
| 5AGV | 758 | 3464 | 6873 | **78.1**/89.0 | 22.6 | 31.8 | 35.1 | 28.9/35.6 |
| 6AGV | 538 | 3405 | 9901 | 84.2/94.6 | 20.2 | 30.7 | 42.2 | 34.2/52.1 |
| 7AGV | 927 | 5601 | 13166 | 83.4/93.0 | 25.2 | 35.3 | 57.1 | 28.6/**55.9** |

Note: Values are rounded to one decimal place. Improved efficiency is calculated as:

As can be seen from Fig 8, the traditional ACO algorithm only uses the path length as the iterative goal, ignoring the time loss caused by the turn, resulting in the shortest path length, but the actual loss of time is the longest phenomenon among the three algorithms. NACO, although it successfully introduces the concept of the function of time, replaces the traditional AGV path length to do the ACO iterative pheromone, making the deviation of the ideal moving time of the AGV from the actual moving time smaller, but the disadvantage is that the time cost of any turning angle is not refined enough, resulting in the lack of path smoothness during the ACO iteration. The deviation between the ideal AGV movement time and the actual movement time becomes smaller, the number of turns is reduced, and the path smoothness is higher, but the disadvantage is that the time cost of the turn angle is not refined enough, and the cost of any turn angle is the same, which leads to the lack of sensitivity to the change of the turn angle in the iteration process of the ant colony. IACO further refines the cost of the turn time on the basis of NACO and introduces the cost of the turn time of the unit angle, which makes the overall AGV path planning time reduced compared with IACO. Time is reduced by 2.66 s compared to the NACO algorithm, and the turn angle is reduced by 90° (Table 5).

As shown in Figs 9 and 10, with the start/end times in Table 3, both NACO and IACO achieve collision-free multi-AGV paths: overlaps are completely time-staggered and all post-planning conflict rates are 0% across 3–7 AGVs.

Since the prioritization of the NACO-based multi-AGV collision-free algorithm is only divided by the order of the input task coordinates, it can further be seen from the path information in Fig 9(a) that, among the three AGVs departing at the

**(a)** ACO **(b)** NACO **(c)** IACO

Fig 8. Simulation of Single AGV path planning with different algorithms.

**Table 5. Path planning post-processing.**

| Algorithm | Performance indicators | | | | | |
|---|---|---|---|---|---|---|
| | Start | End | Ideal time(s) | Actual time(s) | Turn angle(°) | Path length(m) |
| ACO | (1,1) | (18,20) | 60.28 | 102.78 | 765 | 30.14 |
| NACO | (1,1) | (18,20) | 70.28 | 85.28 | 450 | 30.14 |
| IACO | (1,1) | (18,20) | 82.62 | 82.62 | 360 | 31.31 |

Note: Values are rounded to two decimal places. For ACO and NACO, the Ideal time and Actual time are computed as follows:

same time, AGV1 and AGV2, which have a long path time cost, become high priority, while AGV3, which has a short path time cost, becomes the lowest priority, which makes AGV3 collision-free in path planning forced to wait with rerouting. However, due to the short paths between the start and end points of AGV3, there are relatively fewer nodes to choose for replanning, coupled with the fact that NACO is a combination of path planning, waiting strategy, and ant colony pheromone, which triggers the problem that the ant colony is prone to fall into the problem of locally optimal solutions and eventually generate curved paths repeatedly in the local area, making the overall path planning time cost increase. And from the path information in Fig 9(c), it can be seen that the path planning quality of 3AGVs is very good under the IACO-based multi-AGV collision-free strategy because the prioritization of IACO is to do single-AGV path planning for the AGVs running at the same time. This ensures that AGV3, which has a short path and a small time cost, is given priority to complete the task, while AGV1 and AGV2, which have a long path time, can easily find different paths close to the optimal solution due to having more nodes to choose from. Meanwhile, IACO decouples the waiting strategy from the ant colony pheromone, uses the optimal single-AGV initial path to wait for the exact conflict point instead of relying on the decision threshold in the colony pheromone, and finally chooses the strategy by comparing the size of the time cost of global waiting and path replanning, avoiding the colony falling into the problem of local optimality and making the overall quality of multi-AGV collision-free path planning enhancement. As can be seen from Table 6, under simultaneous multi-AGV operation, IACO reduces the total running time relative to NACO by up to 29.59 s and the cumulative turning angle by up to 450°, yielding better path smoothness.

## Conclusion

This paper investigates the multi-AGV collision-free path-planning problem in the context of the textile industry. The primary innovations are as follows: combining the concept of R-tree index with multi-AGV path conflict detection, designing the RLBCD algorithm according to the characteristics of AGV traveling paths, and greatly improving the efficiency of conflict detection among multi-AGVs; the IACO algorithm is proposed to introduce the turn unit angle time factor to participate in the ant colony pheromone updating, which makes the AGV path planning quality better and shortens the driving time; design the same time start AGVs is prioritized according to the initial path time cost and the collision-free path strategy planning is carried out according to the priority, which improves the efficiency of the collision-free path planning for multiple AGVs; to make the AGV path replanning fully take into account the information of the high-priority AGV paths, various types of dynamic obstacle areas are generated according to the refinement of the high-priority AGV path movement mode, which is deeply fused with the IACO algorithm to ensure that the conflict rate of the path replanning is 0; to fully utilize the initial optimal paths and avoid replanning into deadlock problems, the dynamic waiting strategy is used, and the new path time after waiting is compared with the replanning path time, and the path strategy with the shortest time is selected, ensuring time efficiency first.

Through experimental simulation, it is proved that under multiple groups with different numbers of AGVs, the RLBCD algorithm proposed in this paper can effectively reduce the number of conflict detection times more than both CBS and GTCD algorithms, and the comparative times are reduced by at least 78% and 88%, respectively; the proposed IACO

 

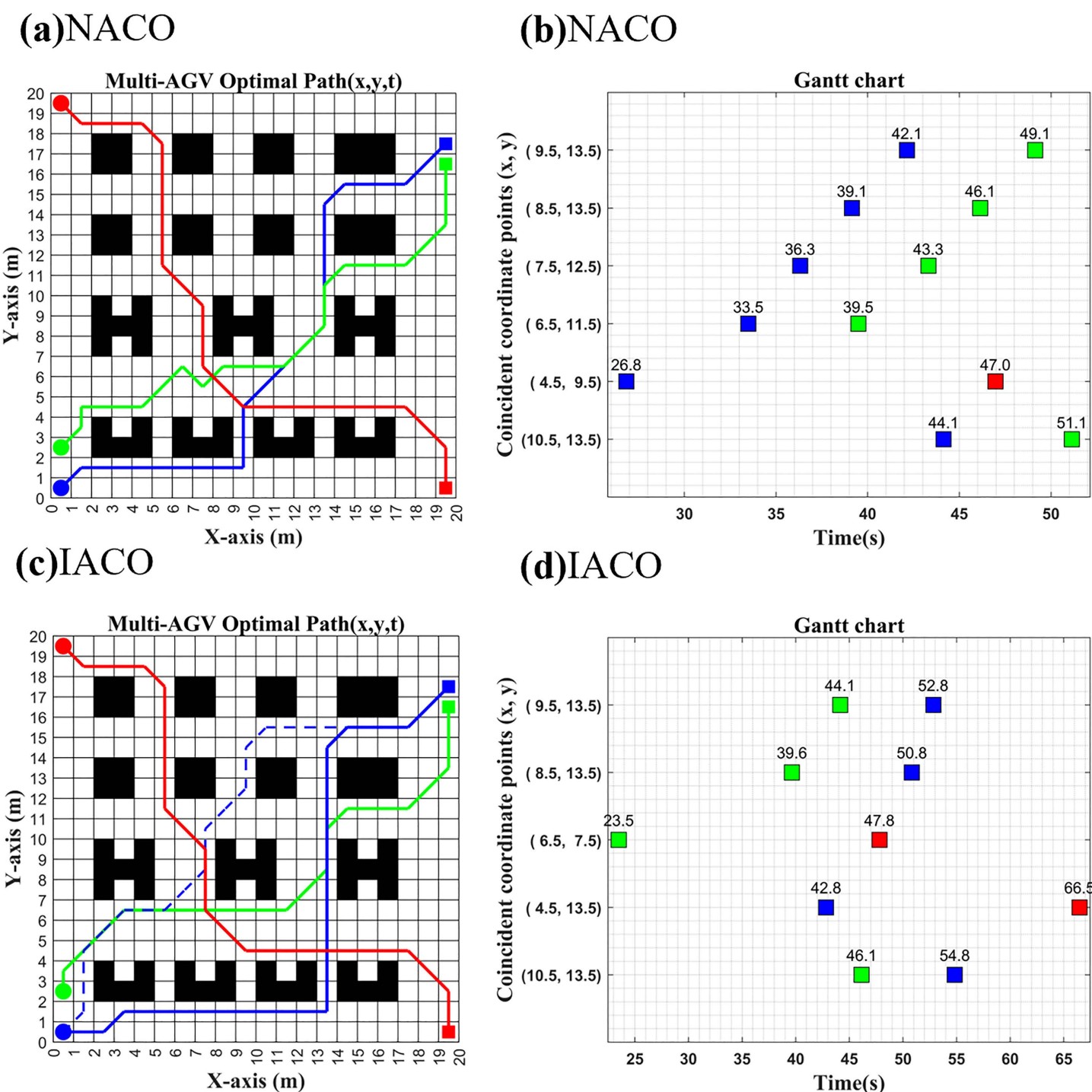

**Fig 9. 3 AGVs started at the same time with different algorithms to plan.**

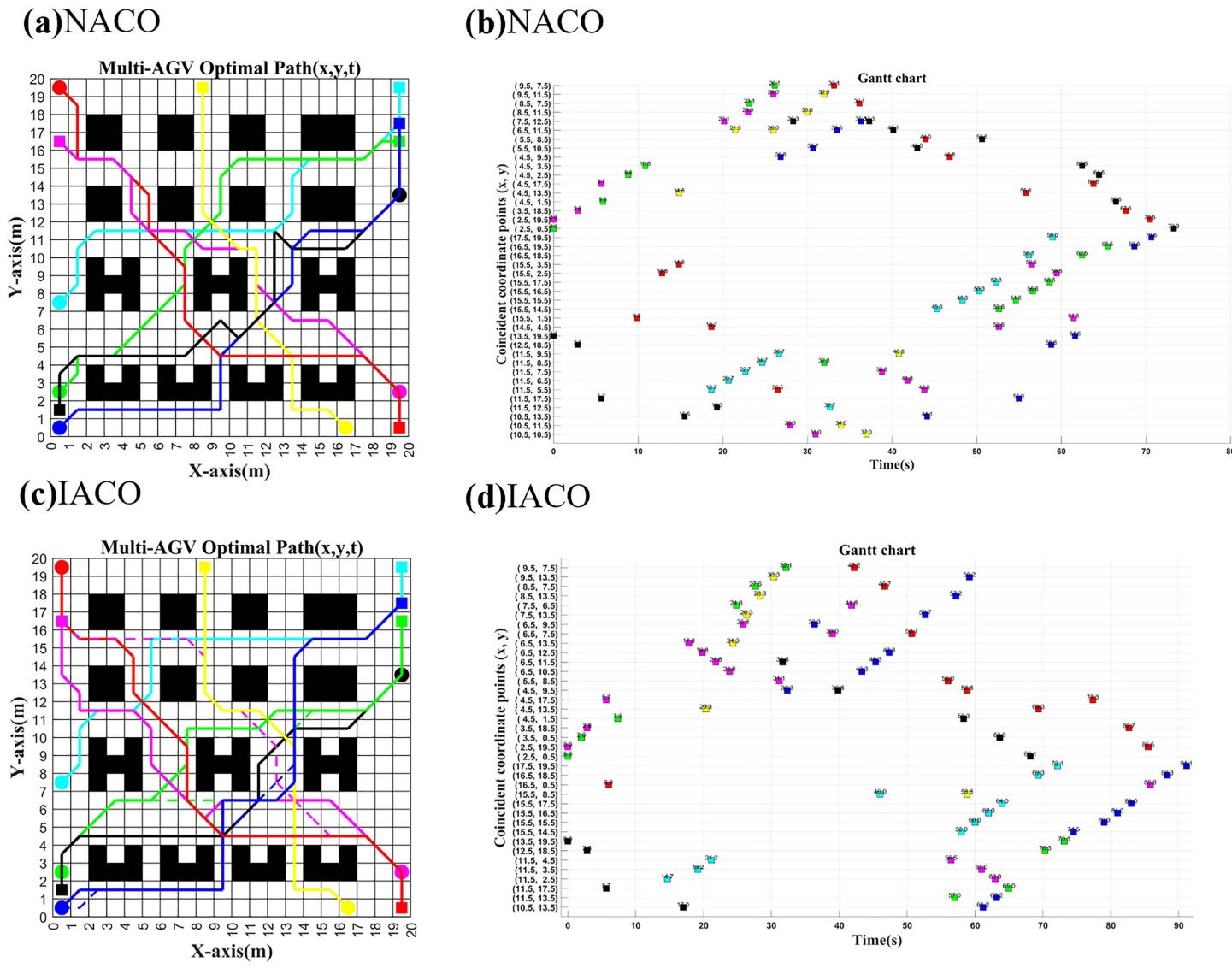

**Fig 10. 7 AGVs started at the same time with different algorithms to plan.**

**Table 6. 3-7 AGVs route planning post-processing.**

| Indicators | NACO/ IACO | | | | NACO- IACO |
|---|---|---|---|---|---|
| | Path length (m) | Total time(s) | Total turn angle(°) | Conflict | m/s/° |
| 3 AGV | 93.77/95.28 | 272.88/250.57 | 1440/1080 | 0/0 | −1.51/22.31/360 |
| 4 AGV | 118.67/120.43 | 339.84/328.35 | 1845/1575 | 0/0 | −1.79/11.49/270 |
| 5 AGV | 144.64/148.15 | 422.78/418.81 | 2385/2205 | 0/0 | **−3.51/3.97/180** |
| 6 AGV | 169.88/172.23 | 502.92/491.95 | 2835/2610 | 0/0 | −2.35/10.97/225 |
| 7 AGV | 199.61/202.30 | 595.71/566.12 | 3330/2880 | 0/0 | −2.69/**29.59/450** |

Note: Values are rounded to two decimal places. NACO (original) uses a fixed 1 s/turn penalty. For cross-method comparability, the corrected NACO total time is recomputed using the same angle-proportional turning loss as IACO, i.e., 10/180 s/°. In addition to turning and path-length costs, the total time also includes waiting-time losses.

not only has a better improvement than NACO in single-AGV path planning but also reduces the total runtime by up to 29.59 s in multi-AGV collision-free path planning and the total turning angle is reduced by up to 450°, yielding better path smoothness. Although the RLBCD algorithm proposed in this paper can improve the conflict detection efficiency in most of the multi-AGV scenarios, it can be seen from the experimental simulation that in the case of a very small number of AGVs, due to the complexity of the data structure, the operation rate of RLBCD is not as good as that of the CBS algorithm, which is a point that can be further improved by further consideration.

## Supporting information

**S1 Dataset. Raw data for Experiment 1.** This file contains the underlying numerical data for Fig 5.
(XLSX)

**S2 Dataset. Raw data for Experiment 2.** This file contains the underlying numerical data for Fig 6.
(XLSX)

**S3 Dataset. Raw data for Experiment 3.** This file contains the underlying numerical data for Figs 8–10.
(XLSX)

## Acknowledgments

We would like to thank the Academic Editor and the anonymous reviewers for their constructive comments and suggestions, which have significantly improved the quality of this manuscript.

## Author contributions

**Conceptualization:** wei Xie, XiangLe Zheng.

**Data curation:** XiangLe Zheng, Jiachen Ma, Jun Chen, Xiaoli Wang.

**Formal analysis:** wei Xie, XiangLe Zheng, Bin Du.

**Funding acquisition:** wei Xie, Jiachen Ma, Jun Chen, Xiaoli Wang.

**Investigation:** wei Xie, XiangLe Zheng, Jun Chen, Bin Du.

**Methodology:** wei Xie, XiangLe Zheng.

**Project administration:** wei Xie, Jiachen Ma, Jun Chen.

**Resources:** wei Xie, Jiachen Ma, Jun Chen, Xiaoli Wang.

**Software:** wei Xie, XiangLe Zheng.

**Supervision:** wei Xie, XiangLe Zheng, Jiachen Ma, Jun Chen.

**Validation:** wei Xie, XiangLe Zheng, Jiachen Ma.

**Visualization:** wei Xie, XiangLe Zheng.

**Writing – original draft:** XiangLe Zheng.

**Writing – review & editing:** wei Xie, XiangLe Zheng, Jiachen Ma, Jun Chen, Bin Du.

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
