## [Decision Letter · Decision Letter 0]

27 Oct 2025

PONE-D-25-27260
Multi-AGV path planning method for the textile industry based on improved ant colony optimization algorithm and R-tree line-segment bilayer conflict detection
PLOS ONE

Dear Dr. Zheng,

Thank you for submitting your manuscript to PLOS ONE. After careful consideration, we feel that it has merit but does not fully meet PLOS ONE’s publication criteria as it currently stands. Therefore, we invite you to submit a revised version of the manuscript that addresses the points raised during the review process.

We look forward to receiving your revised manuscript.

Kind regards,

Reza Rostamzadeh

Academic Editor

PLOS ONE

Journal Requirements:

“This work was supported by the National Key R&D Program of China (No. 2022YFB4700601，W.X.), the National Key R&D Program of China (No.2022YFB4700602,W.X.), the Taishan Scholars (No. tsqn201909153,X.W.), the Ministry of Education industry-university cooperative education project (No. 22086429092517,W.X.).”

“This work was supported by the National Key R&D Program of China (No. 2022YFB4700601，W.X.), the National Key R&D Program of China (No.2022YFB4700602,W.X.), the Taishan Scholars (No. tsqn201909153,X.W.), the Ministry of Education industry-university cooperative education project (No. 22086429092517,W.X.).”

5. We note that your Data Availability Statement is currently as follows: [All relevant data are within the manuscript and its Supporting Information files.]

Reviewers' comments:

Reviewer's Responses to Questions

**Comments to the Author**

1. Is the manuscript technically sound, and do the data support the conclusions?

Reviewer #1: Yes

Reviewer #2: Yes

2. Has the statistical analysis been performed appropriately and rigorously? 

Reviewer #1: Yes

Reviewer #2: Yes

3. Have the authors made all data underlying the findings in their manuscript fully available?

Reviewer #1: No

Reviewer #2: Yes

4. Is the manuscript presented in an intelligible fashion and written in standard English?

Reviewer #1: Yes

Reviewer #2: Yes

5. Review Comments to the Author

Reviewer #1: (1) In the RLBCD model, there are numerous line segments. How do you determine the length of each line segment? What is the basis for dividing the lines into segments? Please provide more detailed information on this aspect.

(2) Eliminating AGV path conflicts and deadlocks is an industry challenge. The literature review on AGV path conflicts in section 2.1 is not comprehensive enough and needs to be supplemented. In fact, there are many works studied this issue. For example, [1] and [2] use time-window based search methods to find conflict free paths, [3] and [4] solve conflicts based on area locking, and [5] eliminates path conflicts from the perspective of adjusting AGV task assignments.

(3) The paper presents an AGV obstacle avoidance strategy that uses waiting and replanning. However, after employing these two methods, conflicts may occur again. How do you address this issue?

(4) You mentioned the use of CBS and GTCD methods in your benchmark. However, you did not give a clear explanation of how these two methods were specifically utilized in your study. To enhance the clarity and reproducibility of your research, it would be beneficial for you to briefly introduce the application of CBS and GTCD in your paper.

(5) After adopting the proposed method, is the conflict rate of AGVs zero for experimental cases of different scales? If not, there should be experiments to demonstrate the improvement of the proposed method in terms of the conflict rate.

References:

[1] Fan, Z., Gu, C., Yin, X., Liu, C., & Huang, H. (2017, December). Time window based path planning of multi-AGVs in logistics center. In 2017 10th International Symposium on Computational Intelligence and Design (ISCID) (Vol. 2, pp. 161-166). IEEE.

[2] Xin, J., Meng, C., Schulte, F., Peng, J., Liu, Y., & Negenborn, R. R. (2020). A time-space network model for collision-free routing of planar motions in a multirobot station. IEEE Transactions on Industrial Informatics, 16(10), 6413-6422.

[3] Kim, K. H., Jeon, S. M., & Ryu, K. R. (2007). Deadlock prevention for automated guided vehicles in automated container terminals. Container Terminals and Cargo Systems: Design, Operations Management, and Logistics Control Issues, 243-263.

[4] Małopolski, W. (2018). A sustainable and conflict-free operation of AGVs in a square topology. Computers & Industrial Engineering, 126, 472-481.

[5] Li, S., Fan, L., & Jia, S. (2024). A hierarchical solution framework for dynamic and conflict-free AGV scheduling in an automated container terminal. Transportation Research Part C: Emerging Technologies, 165, 104724.

Reviewer #2: After reviewing your manuscript on multi-AGV path planning with RLBCD and IACO, I find the work addresses an important problem in textile industry automation and presents interesting ideas for improving conflict detection efficiency through R-tree indexing and refined ant colony optimization. However, I have identified several issues requiring major revision:

1. Missing Theoretical Complexity Analysis

The paper lacks theoretical complexity analysis for the RLBCD algorithm. While experimental runtime comparisons are provided, there's no Big-O notation or formal analysis of how the algorithm scales with increasing numbers of AGVs, path lengths, or map sizes. For a paper proposing a new algorithmic approach, this theoretical foundation is essential for readers to understand the fundamental scaling properties beyond your specific implementation and test cases.

2. Inconsistent Time Calculations in Results (Tables 5 & 6)

There's a problem with the reported times that calls into question the validity of your experimental comparison. In Table 5, IACO shows identical ideal and actual times (82.6s), yet reports 360° of turning. Based on your equations 4-6 and stated parameters (t_turn_unit = 0.055s), this turning should add 19.8 seconds. With a 31.3m path at 0.5 m/s, the movement time alone is 62.6s, so actual time should be approximately 82.4s, not the same as ideal.

This pattern repeats in Table 6 where total ideal equals total actual (250.5s) despite 1080° of cumulative turning across three AGVs. Additionally, NACO's ideal time (70.3s) doesn't match what should be calculated from its path length (60.2s for 30.1m).

These inconsistencies suggest either the "ideal time" is defined differently for different algorithms, there are calculation errors, or the turning time model isn't being applied correctly. This needs to be clarified and corrected for meaningful algorithm comparison.

3. Minor Syntax Issue

Algorithm 1, line 6: "determine θᵢ in using relative direction of neighboring three nodes" - the word "in" appears to be either a typo or incomplete phrase. Should read "determine θᵢ using relative direction..."

The time calculation issue is particularly critical as it directly impacts your main claims about algorithm performance. Please address these concerns in your revision.

6. PLOS authors have the option to publish the peer review history of their article (what does this mean?). If published, this will include your full peer review and any attached files.

Reviewer #1: No

Reviewer #2: No

---

## [Author Response · Author response to Decision Letter 1]

12 Nov 2025

Dear Editor and Reviewers,

Thank you for offering us an opportunity to improve the quality of our submitted manuscript (Article Number: [PONE-D-25-27260] / Title: “[Multi-AGV path planning method for the textile industry based on improved ant colony optimization algorithm and R-tree line-segment bilayer conflict detection]”). We appreciated very much the reviewers’ constructive and insightful comments. In this revision, we have addressed all of these comments/suggestions. We hope the revised manuscript has now met the publication standard of your journal.

Due to formatting issues with the formulas in the response, this window reply only provides a brief explanation of the modifications. For details, please refer to the specific document "Response to Reviewers" and the revised "Manuscript".Below we provide point-by-point responses and indicate where the corresponding revisions were made.

Reviewer #1:

(1) In the RLBCD model, there are numerous line segments. How do you determine the length of each line segment? What is the basis for dividing the lines into segments? Please provide more detailed information on this aspect.

Response:

Thank you for asking for a crisper definition—this question goes to the heart of RLBCD’s correctness and we appreciate your precision. We now make the segmentation rule explicit in RLBCD model: after IACO, each AGV path is partitioned into co-directional segments using turning-coordinate indices; then each segment is subdivided by consecutive node pairs with well-defined time windows. Coarse screening uses (i) time-window overlap and (ii) segment-to-segment minimum distance vs. D_safe; fine screening verifies on subdivided segment units. See Eqs. (13)-(16) (definitions) and Eqs. (17)-(19) (coarse/fine checks).

(2) Eliminating AGV path conflicts and deadlocks is an industry challenge. The literature review on AGV path conflicts in section 2.1 is not comprehensive enough and needs to be supplemented. In fact, there are many works studied this issue. For example, [1] and [2] use time-window based search methods to find conflict free paths, [3] and [4] solve conflicts based on area locking, and [5] eliminates path conflicts from the perspective of adjusting AGV task assignments.

Response：

We are very grateful for your additions. After reading the references you provided, we found that our introduction indeed contains citation gaps. We expanded “Introduction” to cover CBS, time-expanded/time-window networks, regional mutual exclusion (area locking), and task-assignment-driven deconfliction, with the cited works incorporated. We also positioned GTCD (priority-based, edge-level checks) for clear contrast. In addition, at the end of the introduction we have outlined the structure of our paper and the proposed contributions to help readers better grasp our ideas.

(3) The paper presents an AGV obstacle avoidance strategy that uses waiting and replanning. However, after employing these two methods, conflicts may occur again. How do you address this issue?

Response:

Thank you very much for your detailed question. Your insight is excellent. Indeed, for certain algorithms such as CBS, new conflicts may arise even after applying waiting or restarting, necessitating further conflict detection to handle the newly emerging cases.

In our framework, priorities allow us to pre-assign responsibility for deconfliction: high-priority AGVs have the right of way and do not need to consider conflicts with lower-priority AGVs. Accordingly, we map the subdivided path segments of high-priority AGVs into dynamic-obstacle regions for lower-priority AGVs. The nodes of these regions are dynamically inserted into the tabu list of the ant colony algorithm. When the colony approaches such a region, if the time of moving to the next node overlaps with the activation time window of the dynamic obstacle, that node is placed on the tabu list and cannot be selected. Combined with the pheromone positive-feedback mechanism, once the colony successfully finds a path, the new path is guaranteed to be free of conflicts with higher-priority AGVs. By iterating this process, all AGVs can operate stably without collisions.

For the waiting strategy, it is true that the planned path must be rechecked; however, because priorities are in place, a lower-priority AGV only needs to rerun RLBCD, which quickly identifies the next set of segments that conflict with higher-priority AGVs and thus enables rapid waiting decisions.

We also considered the extreme case you mentioned and added a supplementary safeguard: if the ant colony algorithm still fails within the allowed number of replanning attempts—i.e., at some time instant no feasible solution can be found due to dense dynamic obstacles—and the waiting strategy exceeds its maximum time limit, the current AGV’s task is canceled and deferred to the next departure batch. This measure ensures that already planned high-priority paths remain unaffected. We have incorporated this point in both the introduction and the strategy-selection section of the paper.

(4) You mentioned the use of CBS and GTCD methods in your benchmark. However, you did not give a clear explanation of how these two methods were specifically utilized in your study. To enhance the clarity and reproducibility of your research, it would be beneficial for you to briefly introduce the application of CBS and GTCD in your paper.

Response:

Thank you for emphasizing reproducibility—we agree that baseline configurations should be transparent. In “Experimental Results and Analysis”, we added a concise paragraph: three methods share the same (x, y, t) inputs and the same conflict criterion (distance below\ D_{\mathrm{safe}}\ and linear time-window overlap). GTCD applies our priority rule then performs direct edge-level checks; RLBCD encapsulates into co-directional segments with coarse-to-fine screening; CBS uses point/edge checks without priority. The detection count is defined as one full spatiotemporal check per path. We have also provided a more detailed description of GTCD and CBS in the “Introduction”.

(5) After adopting the proposed method, is the conflict rate of AGVs zero for experimental cases of different scales? If not, there should be experiments to demonstrate the improvement of the proposed method in terms of the conflict rate.

Response:

We appreciate the request for scale-sensitivity evidence. To further verify, at the experimental scales considered (3-7 AGVs), whether the algorithm enables collision-free simulation across different fleet sizes, we added supplementary experiments. In particular, we include the actual simulation snapshots and the corresponding Gantt chart for the 7-AGV case to demonstrate the absence of conflicts (Fig10.), and we summarize the results for each fleet size in Table 6. These additions further clarify, under identical experimental scales, parameter settings, and attempt counts, the differences and relative merits between our proposed algorithm and a NACO-based ant-colony deconfliction strategy. To enable more rigorous data validation, we will provide, in addition to the data reported in the paper, detailed path-node records for all experiments, including computed path lengths, cumulative turning angles, and per-AGV execution details before and after deconfliction following plan acceptance.

Reviewer #2: After reviewing your manuscript on multi-AGV path planning with RLBCD and IACO, I find the work addresses an important problem in textile industry automation and presents interesting ideas for improving conflict detection efficiency through R-tree indexing and refined ant colony optimization. However, I have identified several issues requiring major revision:

1. Missing Theoretical Complexity Analysis

The paper lacks theoretical complexity analysis for the RLBCD algorithm. While experimental runtime comparisons are provided, there's no Big-O notation or formal analysis of how the algorithm scales with increasing numbers of AGVs, path lengths, or map sizes. For a paper proposing a new algorithmic approach, this theoretical foundation is essential for readers to understand the fundamental scaling properties beyond your specific implementation and test cases.

Response:

We are deeply honored by your recognition of the innovations in our work. Your positive assessment is both encouraging and motivating. Thank you for insisting on theoretical scalability—this is essential for an algorithmic paper. We added a dedicated subsection, “RLBCD time-complexity,” presenting Big-O bounds for preparation, coarse screening, fine screening, per-segment, and per-AGV costs (Eqs. (20)-(25)), and analyzing the worst-case scenario.

2. Inconsistent Time Calculations in Results (Tables 5 & 6)

There's a problem with the reported times that calls into question the validity of your experimental comparison. In Table 5, IACO shows identical ideal and actual times (82.6s), yet reports 360° of turning. Based on your equations 4-6 and stated parameters (t_turn_unit = 0.055s), this turning should add 19.8 seconds. With a 31.3m path at 0.5 m/s, the movement time alone is 62.6s, so actual time should be approximately 82.4s, not the same as ideal.

This pattern repeats in Table 6 where total ideal equals total actual (250.5s) despite 1080° of cumulative turning across three AGVs. Additionally, NACO's ideal time (70.3s) doesn't match what should be calculated from its path length (60.2s for 30.1m).

These inconsistencies suggest either the "ideal time" is defined differently for different algorithms, there are calculation errors, or the turning time model isn't being applied correctly. This needs to be clarified and corrected for meaningful algorithm comparison.

Response:

Thank you for your insightful comments; you are correct that our calculations contained errors. Upon careful verification, we identified three sources. First, path lengths were manually truncated to one decimal place (e.g., 30.1 m instead of 30.14 m), which led the code to round the computed time, e.g., 30.14x2=60.28≈60.3s. Second, for the IACO unit turning time (0.055), we made a unit-definition error (it should be s/°) and, in the code, we derived this value from (10s/180°), i.e., (0.0555...s/°), but mistakenly stored it as 0.055 without rounding. This yielded 0.055\times180=9.9s, a 0.1s discrepancy, which becomes 0.2s at 360°. Third, whether based on NACO or IACO, the multi-AGV deconfliction algorithm includes a waiting strategy. Consequently, the timing results for the multi-AGV experiments cannot be inferred solely by back-calculating from turning time and path length, which led to a misunderstanding. This was our oversight, as the paper did not provide a detailed explanation. Accordingly, we made the following revisions: we now uniformly round all computed quantities to two decimal places and refine the definition of t_turn_unit^{IACO} to 10/180 s/°. In addition, prompted by your observation, we realized that we had not documented the NACO experimental parameters, which could mislead readers; we therefore added t_turn_unit^{NACO} to Table 1 and clarified the configuration in experiment (iii). To enable more rigorous data validation, we will provide, in addition to the data reported in the paper, detailed path-node records for all experiments, including computed path lengths, cumulative turning angles, and per-AGV execution details before and after deconfliction following plan acceptance.

3. Minor Syntax Issue

Algorithm 1, line 6: "determine θᵢ in using relative direction of neighboring three nodes" - the word "in" appears to be either a typo or incomplete phrase. Should read "determine θᵢ using relative direction..."

Response:

Thank you very much for pointing out the grammatical errors; we have made the corresponding revisions.

We sincerely appreciate the reviewers’ deep insights, which prompted us to unify the timing model and rounding, clarify baseline configurations, strengthen theoretical analysis, and make the deconfliction logic more explicit. Thank you again for your time and consideration.

---

## [Decision Letter · Decision Letter 1]

9 Dec 2025

Multi-AGV path planning method for the textile industry based on improved ant colony optimization algorithm and R-tree line-segment bilayer conflict detection

PONE-D-25-27260R1

Dear Dr. Zheng,

We’re pleased to inform you that your manuscript has been judged scientifically suitable for publication and will be formally accepted for publication once it meets all outstanding technical requirements.

Kind regards,

Reza Rostamzadeh

Academic Editor

PLOS One

Additional Editor Comments (optional):

Reviewers' comments:

Reviewer's Responses to Questions

**Comments to the Author**

1. If the authors have adequately addressed your comments raised in a previous round of review and you feel that this manuscript is now acceptable for publication, you may indicate that here to bypass the “Comments to the Author” section, enter your conflict of interest statement in the “Confidential to Editor” section, and submit your "Accept" recommendation.

Reviewer #1: All comments have been addressed

Reviewer #2: All comments have been addressed

2. Is the manuscript technically sound, and do the data support the conclusions?

Reviewer #1: Yes

Reviewer #2: Yes

3. Has the statistical analysis been performed appropriately and rigorously? 

Reviewer #1: Yes

Reviewer #2: Yes

4. Have the authors made all data underlying the findings in their manuscript fully available?

Reviewer #1: (No Response)

Reviewer #2: Yes

5. Is the manuscript presented in an intelligible fashion and written in standard English?

Reviewer #1: (No Response)

Reviewer #2: Yes

6. Review Comments to the Author

Reviewer #1: The authors have adequately addressed all my comments. The paper quality has now been significantly improved and it is suitable for publication.

Reviewer #2: The revision successfully addresses ALL reviewer comments. The authors have been thorough in their responses, adding substantial new content including complexity analysis, expanded literature review, explicit methodology for calculations, and additional experimental validation across fleet sizes.

Accept with minor revisions:

1. Clarify the R-tree terminology (either implement actual R-tree or rename to "hierarchical bilayer screening")

2. Add a brief discussion of scalability limitations and when CBS might be preferred (small fleets)

7. PLOS authors have the option to publish the peer review history of their article (what does this mean?). If published, this will include your full peer review and any attached files.

Reviewer #1: No

Reviewer #2: No

---

## [Editor Report · Acceptance letter]

PONE-D-25-27260R1

PLOS One

Dear Dr. Zheng,

I'm pleased to inform you that your manuscript has been deemed suitable for publication in PLOS One. Congratulations! Your manuscript is now being handed over to our production team.

Kind regards,

on behalf of

Dr. Reza Rostamzadeh

Academic Editor

PLOS One